# Exploring the interplay between the core microbiota, physicochemical factors, agrobiochemical cycles in the soil of the historic tokaj mád wine region

Judit Remenyik[1]☯, László Csige[2]☯, Péter Dávid[1], Péter Fauszt[1], Anna Anita Szilágyi-Rácz[1], Erzsébet Szőllősi[1], Zsófia Réka Bacsó[2], István Szepsy Jnr[2], Krisztina Molnár[3], Csaba Rácz[3], Gábor Fidler[1], Zoltán Kállai[4], László Stündl[5], Attila Csaba Dobos[3‡], Melinda Paholcsek[1‡]*

1 Center for Complex Systems and Microbiome Innovations, Faculty of Agricultural and Food Sciences and Environmental Management, University of Debrecen, Debrecen, Hungary, 2 Research Laboratory and Wine Academy of Mad, University of Debrecen, Mád, Hungary, 3 Centre for Precision Farming R&D Services, Faculty of Agriculture, Food Science and Environmental Management, University of Debrecen, Debrecen, Hungary, 4 Department of Genetics and Applied Microbiology, Faculty of Science and Technology, University of Debrecen, Debrecen, Hungary, 5 Institute of Food Technology, Faculty of Agricultural and Food Sciences and Environmental Management, University of Debrecen, Debrecen, Hungary

☯ These authors contributed equally to this work.
‡ MP and ACD also contributed equally to this work.
* paholcsek.melinda@agr.unideb.hu

**Data Availability Statement:** All sequencing data are available from the Sequence read Archive (SRA) (http://www.ncbi.nlm.nih.gov/sra) under

## Abstract

A Hungarian survey of Tokaj-Mád vineyards was conducted. Shotgun metabarcoding was applied to decipher the microbial-terroir. The results of 60 soil samples showed that there were three dominant fungal phyla, *Ascomycota* 66.36% ± 15.26%, *Basidiomycota* 18.78% ± 14.90%, *Mucoromycota* 11.89% ± 8.99%, representing 97% of operational taxonomic units (OTUs). Mutual interactions between microbiota diversity and soil physicochemical parameters were revealed. Principal component analysis showed descriptive clustering patterns of microbial taxonomy and resistance gene profiles in the case of the four historic vineyards (Szent Tamás, Király, Betsek, Nyúlászó). Linear discriminant analysis effect size was performed, revealing pronounced shifts in community taxonomy based on soil physicochemical properties. Twelve clades exhibited the most significant shifts (LDA > 4.0), including the phyla *Verrucomicrobia*, *Bacteroidetes*, *Chloroflexi*, and *Rokubacteria*, the classes *Acidobacteria*, *Deltaproteobacteria*, *Gemmatimonadetes*, and *Betaproteobacteria*, the order *Sphingomonadales*, *Hypomicrobiales*, as well as the family *Sphingomonadaceae* and the genus *Sphingomonas*. Three out of the four historic vineyards exhibited the highest occurrences of the bacterial genus *Bradyrhizobium*, known for its positive influence on plant development and physiology through the secretion of steroid phytohormones. During ripening, the taxonomical composition of the soil fungal microbiota clustered into distinct groups depending on altitude, differences that were not reflected in bacteriomes. Network analyses were performed to unravel changes in fungal interactiomes when comparing postveraison and preharvest samples. In addition to the arbuscular mycorrhiza *Glomeraceae*, the families

PRJNA909960 accession number. Link: https://dataview.ncbi.nlm.nih.gov/object/PRJNA909960?reviewer=de9gvqaubov0ud38dbbtb284al.

**Funding:** The author(s) received no specific funding for this work.

**Competing interests:** The authors have declared that no competing interests exist.

*Mycosphaerellacae* and *Rhyzopodaceae* and the class *Agaricomycetes* were found to have important roles in maintaining soil microbial community resilience. Functional metagenomics showed that the soil Na content stimulated several of the microbiota-related agrobiogeochemical cycles, such as nitrogen and sulphur metabolism; steroid, bisphenol, toluene, dioxin and atrazine degradation and the synthesis of folate.

## Introduction

The consequences of climate change pose challenges to the wine sector worldwide. Wine quality and style are considered to be impacted by the place where the vines grow. Wine minerality is no longer identified with the bedrock but rather with the fungal and bacterial communities of the soil [1, 2]. The relationship between the sensory attributes of the wine and its origin is referred to as "terroir".

The natural conditions and centuries of the wine tradition of Tokaj are so unique that it was named a United Nations Educational, Scientific and Cultural Organization (UNESCO) World Heritage Site in 2002. Wine Region is shaped to a large extent by the region's two rivers, along with their wetlands and moors that make a major contribution to weather conditions, enabling the reliable occurrence of *Botrytis cinerea* in the fall. Other key factors are the terrain and the colourful palette of bedrocks, most of which are volcanic in origin. Mád is one of the 27 settlements of the Tokaj wine region that produces the world-famous aszú wine.

In Tokaj-Mád wine region four historic vineyards (Nyúlászó, Szent Tamás, Király, Betsek) are known to have high historical significance providing first class, prestigious wines praised for their complexity, balance, and distinctive flavour profiles.

*Vitis vinifera* represents an economically and culturally important agricultural crop for which microbial activity plays a critical role in the growth and ripening of grapes [3, 4]. Geology, topography, micro- and mezoclimatic factors, the physical and chemical properties of the soil, grape variety, and genetics such as viticulture practices contribute to the soil microbiota. There is a mutual relationship between a grapevine and its soil microbiota. Microbes communicate through chemical molecules that determine the agrobiochemical processes and the development of the plant. Plants also secrete root exudates, by which they stimulate the soil rhizospheric microbiota [5]. Metabolites produced by the rhizobiome are important contributors to the quality, fragrance, and distinctive quality of the products [6, 7].

Microbial communities are self-sustaining systems in which members continuously compete with each other for the same resources. The adaptation abilities of the microbiota depend on community plasticity. Network analysis-based approaches can help understand the intricate nature of microbe-microbe and microbe-community interactions.

Healthy plants depend on optimal, balanced soil microbiota. As soil harbors both pathogenic and beneficial microorganisms, a better understanding of the community dynamics of the core microbiota might be essential for better decision-making in vitiviniculture practices.

The increase in data storage capacity and the availability of computational algorithms provide the opportunity to unravel community dynamics through next-generation sequencing (NGS) techniques.

Knowledge of the hidden descriptive microbial patterns might provide information about the physical, chemical, and biological functioning of the bulk soil with regard to the metabolic activities linked to the bacteriomes and rhizobiomes. It is a major challenge for researchers to clarify how fungi and bacteria in soil amend terroir characteristics. In the near future, a more

detailed understanding of the microbial community dynamics may open a new horizon of vitiviniculture.

Soil is accounting for approximately 30% of the known antibiotic resistance genes (ARGs) and having a great impact on food security and human health [8–10]. However, the importance of the spatial variability of soil resistomes has not been well explored. Soil ARG profiles also depend on the microbial community structure and functional potential.

As part of our study, we aimed to characterize the microbial landscape of ten vineyards in the renowned Hungarian wine-producing region of Tokaj-Mád including four historic vineyards to map the soil-food web and the 100% core microbiome across three soil levels. Descriptive microbial studies mirror the combined effects of all physical and chemical factors enhancing our understanding of the factors influencing plant health and development. Our goal was to explore the specific microbial clades contributing to the variations among six clusters, which were differentiated based on the physicochemical characteristics of soil samples. We investigated whether soil organic and mineral content correlates more strongly with bacterial diversity than fungal diversity. We also aimed to explore the variations in the 100% bacterial and fungal microbiota of the soil core throughout the ripening stage. This process enables us to capture the cumulative effects of factors such as temperature, drought, rain, grape cultivation practices (including weeds or cover plants, etc.), and plant physiology. We focused on identifying and analyzing the taxonomic shifts within the fungal microbial interactome during the ripening of grapevines. We also aimed to investigate the relationship between bacterial and fungal phylogenetic diversity-, microbial agrobiochemical processes and soil physichochemical parameters of the vineyard soil at varying soil depths.

## Results

### General description of sequencing results

Proper assembly, binning, and taxonomic assignment of metagenomic sequence data are crucial for downstream analyses. After clustering sequences, we obtained 18,162 and 2,064 operational taxonomic units (OTUs) for bacteria and fungi, respectively. In addition, 857 virus and 4,416 nonfungal eukaryote OTUs were identified. On average, 7,900 ± 1,045 bacterial OTUs and 780 ± 234 fungal OTUs were detected in one of the six clusters (SPP1-SPP6) formed by the soil physicochemical properties, with a range of 3,645–9,655 for bacteria and 307 to 1,385 for fungi. A total of 4 kingdoms, 242 phyla, 316 classes, 830 orders, 1,789 families, 6,171 genera, and 16,151 species were detected. A total of 26,666,965 ± 9,001,332 reads were detected as *Bacteria*, 306,217 ± 288,550 reads as *Archaea*, 215,921 ± 889,256 reads as *Eukaryotes*, and 4,041 ± 6,253 reads as *Virus* per sample. A total of 19,865 genes were detected in the KEGG database, while for the analysis, 3,073 metabolism genes were investigated. In our sequencing data, 1,006 antibiotic resistance genes were detected.

### Six clusters were identified based on the physicochemical properties of the bulk soil samples

A hierarchically clustered heatmap was constructed based on the organic and mineral contents in the case of the 42 bulk soil samples of the Mád vineyards representing distinct soil heterogeneity (S1 Fig and S1 Table). The main soil types of all vineyards are Luvisols (brown forest soils with clay illuviation) and Fluvisols (meadow soils with salt accumulation in deeper layers) assigned to clay loam or clay textural classes.

There were six distinct clusters identified according to the soil physicochemical properties (SPP): cluster SPP1 –SPP6. The satellite map represents the 10 vineyards; Vy1: Ősz-hegy, Vy2:

Sarkad, Vy3: Hold-völgy, Vy4: Úrágya, Vy5: Danczka, Vy6: Nyúlászó (historic), Vy7: Szent Tamás (historic), Vy8: Király (historic), Vy9: Betsek (historic), Vy10: Szilvás (Fig 1A). Considering the physicochemical properties of the soil samples Betsek was shown the most heterogeneous, including clusters SPP1-SPP4, followed by vineyard Király, corresponding to two clusters (clusters SPP4 and SPP6) (Fig 1B). Szent Tamás belonged to cluster SPP5. Interestingly, all of the other vineyards were represented by cluster SPP6. Principal coordinate analysis (PCoA) was performed to investigate clustering according to the soil physicochemical properties (Fig 1C) and possible correspondence between the microbial taxonomy of the historical vineyards and the other (Fig 1D). Although there were no sharp differences in the clustering patterns, this did not exclude any codependence between soil physicochemical properties and the soil microbiota. Further integration of microbial taxonomy in the PCoA graph also suggests differences in clustering patterns between the four historic vineyards and the vineyards in their vicinity (Fig 1D). Soil microbiota comparisons were performed among samples on the basis of the most abundant bacterial (Fig 1E) and fungal community compositions (Fig 1F). Remarkable inter-vineyard alterations were observed in the 20 most abundant fungal genera especially in the case of Szent Tamás, Betsek and Sarkad. Szent Tamás showed the highest abundances for the member of the bacterial genus *Bradyrhizobium* which are beneficial to plant development and physiology due to the excretion of the polyhydroxylated steroid phytohormone (e.g. brassinosteroids) production and the lowest frequencies for the genus *Aspergillus* [11]. The root colonizing rhizobacteria *Streptomyces* conferring pathogen resistance dominated in Nyúlászó and Szilvás.

## Pronounced community taxonomy shifts according to soil physicochemical properties

To decipher key bacteria and fungi representing significant shifts in soil elemental composition, the differentially abundant linear discriminant analysis (LDA) effect size (LEfSe) method was used to perform class comparisons among samples of different clusters (cluster SPP1 – SPP6) (Fig 2). We found 12 clades that were the most significantly shifted (LDA>4.0). The phyla *Verrucomicrobia* and *Bacteroidetes*, the order *Sphingomonadales*, the classes *Gemmatimonadetes* and *Betaproteobacteria*, the family *Sphingomonadaceae* and the genus *Sphingomonas* changed significantly in cluster SPP6 (Fig 2A). Two additional clades (c_*Deltaproteobacteria* and p_*Rokubacteria*) showed significant changes in cluster SPP3, and another three clades (p_*Chloroflexi*, o_*Hypomicrobiales* and c_*Acidobacteria*) showed significant changes in cluster SPP2. It was shown that in terms of clades, the largest differences appeared among clusters SPP2, SPP3 and SPP6. Based on the relative frequency data, the phylum *Chloroflexi* and the order *Hypomicrobiales* showed the highest relative abundances in cluster SPP5 (6.85% ± 1.98%, 6.85% ± 0.76%) (Fig 2B).

## Mutual interactions between soil diversity and physicochemical parameters

The correlation between the diversity of soil bacterial and fungal biomes and their organic and mineral matter was investigated. In general, the physico-chemical parameters of soils can either positively or negatively affect microbial communities. (Fig 3). Interestingly, low correlations were observed for fungal community diversity (correlation between soil pH and bacterial Shannon r: 0.55 vs. correlation between soil pH and fungal Shannon r: 0.14) and the physicochemical parameters of the soil (Fig 3A). Regarding the manganese (Mn) and magnesium (Mg) contents, the Spearman correlation resulted in consistently moderate correlations (KCl-soluble Mg mean r: 0.44, plant available Mn mean r: 0.43) for bacteria. In the case of soil bacterial diversity, additional moderate Spearman's correlation coefficients accounted for organic

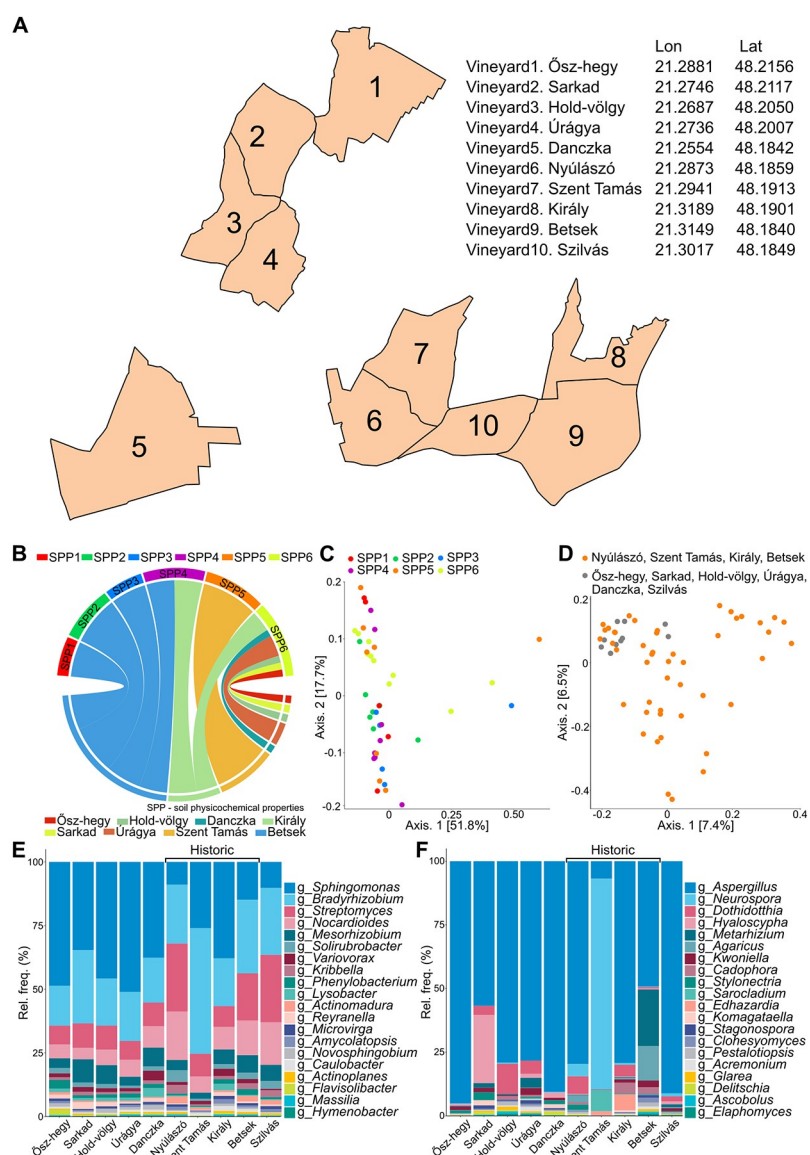

**Fig 1. Codependence of the soil organic and mineral matter and microbial taxonomy of the Tokaj-Mád vineyards.**
**A)** A schematic representation, including GPS coordinates and featuring area-proportional outlines of the vineyards
(Vy1: Ősz-hegy, Vy2: Sarkad, Vy3: Hold-völgy, Vy4: Úrágya, Vy5: Danczka, Vy6: Nyúlászó, Vy7: Szent Tamás, Vy8:
Király, Vy9: Betsek, Vy10: Szilvás). **B)** Chord diagram representing the soil physicochemical properties of the
vineyards. Color codes show the six distinct clusters identified according to the soil physicochemical properties
(SPP1-SPP6) and the vineyards (Ősz-Hegy, Hold-Völgy, Danzka, Király, Sarkad, Úrágya, Szent Tamás, Betsek).
Principal coordinate analysis (PCoA) according to the soil physicochemical properties **C)** and to the normalized
shotgun metagenome sequencing data of the vineyards **D)** based on Bray–Curtis distances. Characterizing the intra-
vineyard variation concerning the 20 most abundant **E)** bacterial and **F)** fungal genera of the Tokaj-Mád.

matter (mean r: 0.3), CaCO$_3$ (mean r: 0.33), Al-soluble P$_2$O$_5$ (mean r: 0.3), and plant available
Co (mean r: 0.35). Interestingly, for plant-available Al, opposite correlation trends were
observed for bacterial vs. fungal alpha diversity (bacterial mean r: -0.27, fungal mean r: 0.26).
Further analysis was conducted to investigate the relationship between soil mineral matter and
bacterial families and genera living in the Mád vineyards. To understand the relationships

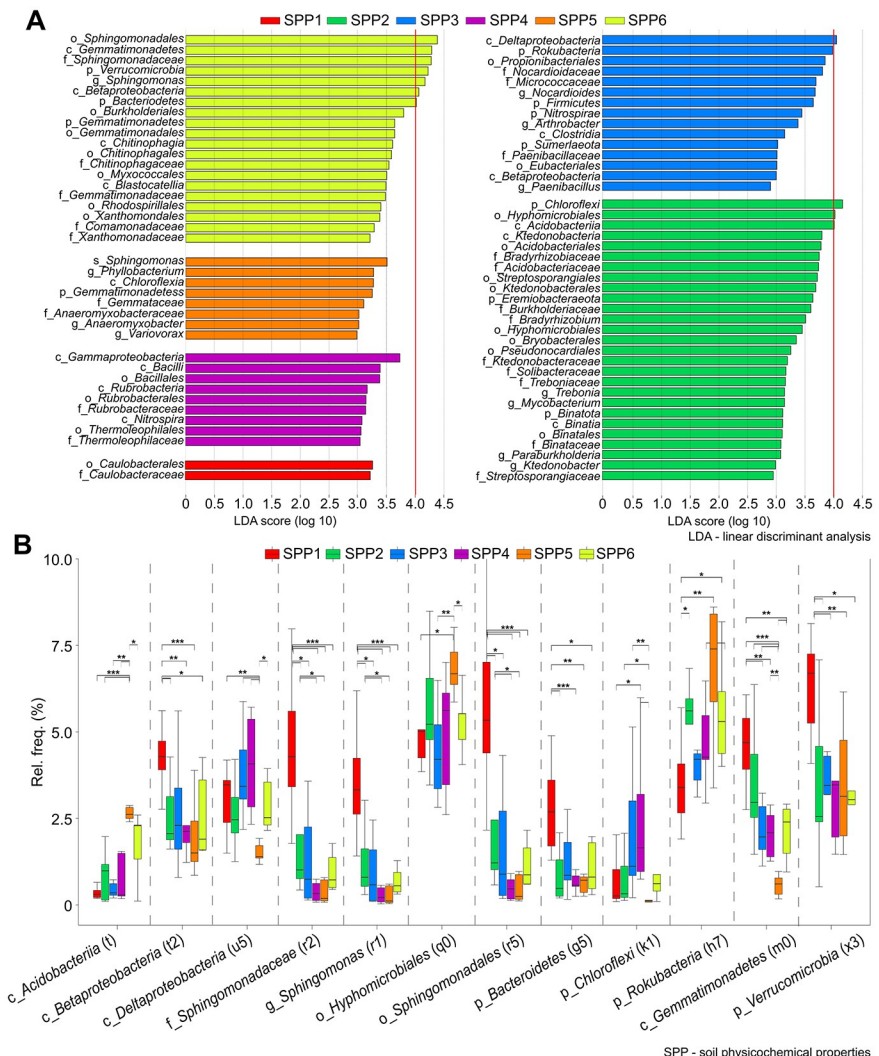

**Fig 2. Linear discriminant analysis effect size (LEfSe) identified microbial clades involved in significant taxonomic shifts. A)** The bar plot depicts the microbial lineages in clusters of soil physicochemical properties (cluster SPP1—SPP6). **B)** A list of 12 bacterial clades, significantly enriched (P value < 0.05), with linear discriminant analysis scores higher than 4 (LDA > 4.0) includes four phyla *Verrucomicrobia*, *Bacteroidetes*, *Rokubacteria*, *Chloroflexi*, two orders *Shingomonadales*, *Hypomicrobiales*, four classes *Gemmatimonadetes*, *Betaproteobacteria*, *Acidobacteria*, *Deltaproteobacteria*, one family *Shingomonadaceae* and one genus *Shingomonas*.

between two sets of variables, canonical correlation analysis (CCA) was used (Fig 3B). CCA is a classic way to evaluate the multivariate associations between two types of high-dimensional data (taxonomy data retrieved from next-generation techniques and soil organic and mineral matter) using canonical vectors. We explored the relationship between two datasets of variables. The family *Sphingomonadaceae* showed associations with plant available sulphur (S), zinc (Zn) plant available copper (Cu), iron (Fe), KCl and soluble sulphate SO4. Further associations were found between plant available calcium (Ca) and *Chitinophagaceae* and *Hymenobacteraceae*. Plant available Al and Na correlated with *Xanthobacteraceae*, *Myxococcaceae*, *Phycispaeraceae*, and *Chtoniobacteraceae*. The relative abundances of *Longimicrobiaceae*, *Erythrobacteriaceae*, and *Sphingosinicellaceae* were correlated with soil organic matter, calcium

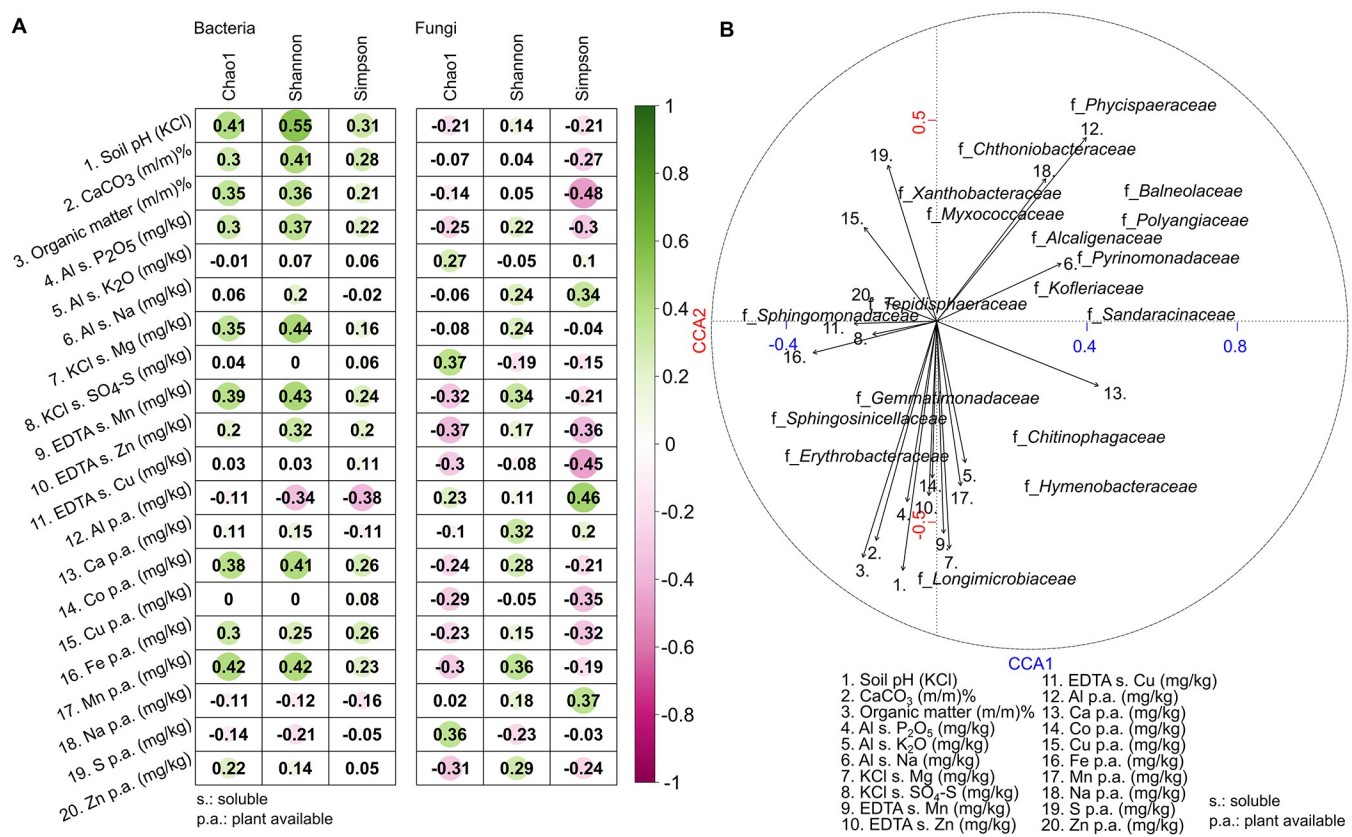

**Fig 3. The organic and mineral matter of soil correlated more strongly with bacterial than fungal diversity. A)** Correlation analysis between soil organic and mineral matter and the bacterial and fungal community diversities. Correlation plots describe the relationships between soil organic and mineral matter and community alpha diversity (ChaoI, Shannon, Simpson). Circles with a darker hue of green demonstrate a stronger positive correlation, and circles with a darker hue of magenta demonstrate a stronger negative correlation. **B)** Shown are the results from canonical correspondence analysis (CCA) representing the ordination of families along the first two axes and their correlation with soil organic and mineral matter.

carbonate ($CaCO_3$), potassium oxide ($K_2O$), magnesium (Mg), manganese (Mn), zinc (Zn), and cobalt (Co) contents.

## The soil food web of the Tokaj-Mád vineyards

The components of the soil-food web (SFW) of the Tokaj-Mád vineyards were also deciphered (Fig 4A). On average, *Bacteria* (98.61% ± 5.43%) and *Archaea* (1.20% ± 1.02%) dominated in the samples. *Virus* (0.010% ± 0.022%) were less frequent than *Fungi* (0.15% ± 0.21%), *Algae* (0.021% ± 0.08%). *Protists* (0.007% ± 0.0021%) were the less frequent component. The relative ratio of DNA derived from *Earthworms* was 0.00066% ± 0.00032%, followed by *Collembola* at 0.00055% ± 0.0016%, *Nematoda* at 0.0019% ± 0.0009% and *Mole* at 5.52E-7% ± 5.53E-7%. By characterizing SFW contents, cluster SPP5 and SPP6 showed the highest relative frequencies for bacteria (SPP5: 99.44%, SPP6: 99.28%) and the lowest for *Fungi* (SPP5: 0.06%, SPP6: 0.066%) and *Archaea* (SPP5: 0.48%, SPP6: 0.62%). *Fungi* peaked in cluster SPP2 (0.42%), *Archaea* peaked in cluster SPP4 (1.70%), and *Algeae* dominated in cluster SPP3 and SPP4 (SPP3: 0.03%, SPP4: 0.04%). Sequencing reads of the kingdom *Bacteria* were the most abundant across all of the samples (Fig 4B). At the phylum level, 70% and 90% of Fungi OTUs and Bacteria OTUs were present in our sample population, respectively. At lower taxonomy levels,

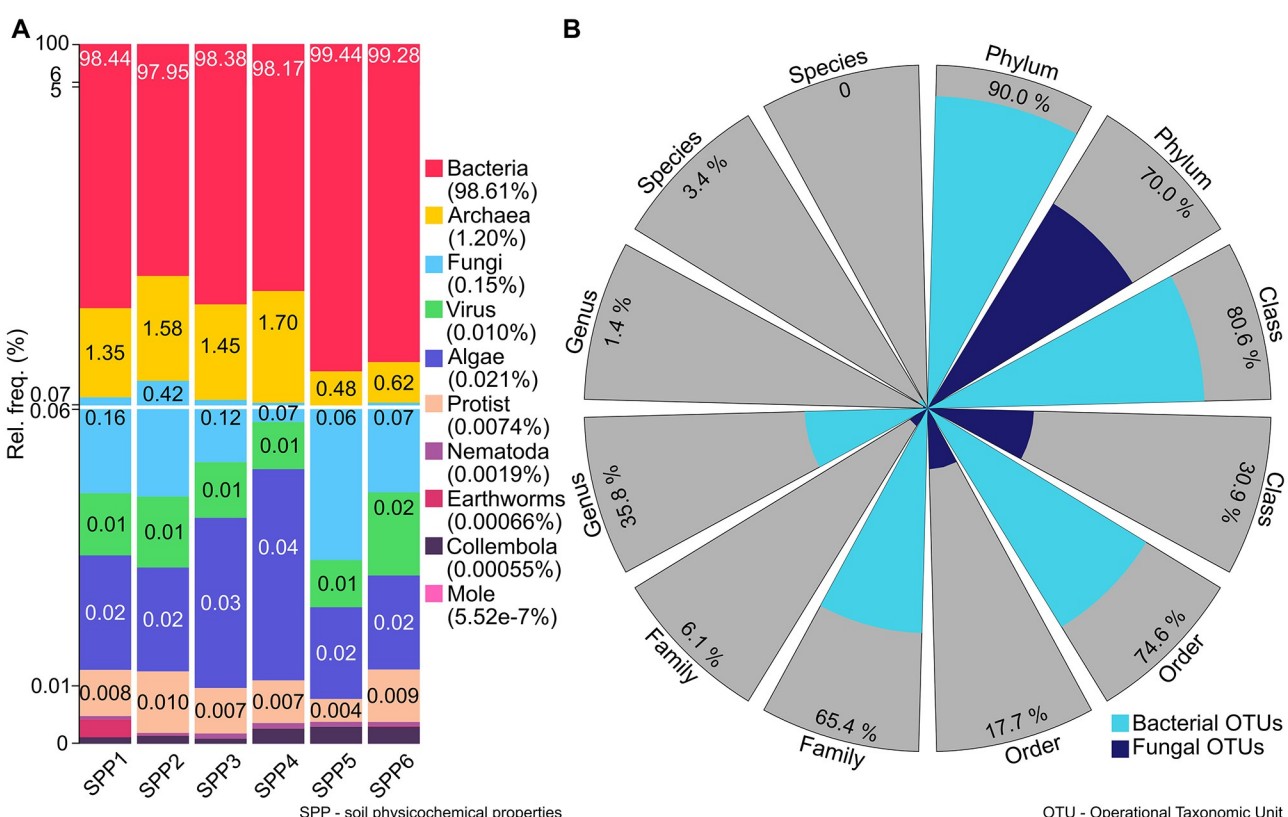

**Fig 4. The soil-food web of the Mád vineyards. A)** Bar plots showing the distribution of the soil food web components of six groups clustered according to the chemical compositions of the soil samples. **B)** Spider-web shows the distribution of the bacterial and fungal OTUs at different taxonomic levels.

these differences tended to increase so that up to the genus level, the number of the bacterial OTUs were shown 2.6 times (class), 4.2 times (order), 10.7 times (family) and 25.6 times (genus) higher than that of the fungal OTUs.

### The 100% core soil microbiota

Comprehensive comparisons of the soil samples of the Tokaj-Mád vineyards were made based on the hierarchical clustering of the bacterial and fungal's 100% core microbiota (Fig 5). A schematized description provides a detailed look at the soil core microbiota of the Mád grape growing area with the following parameters working from the inside out: I. time of sampling (preharvest—postveraison), II. vineyard altitude (140–190m; 191-240m; 241-290m), III. sampling depth (0-30cm; 31-60cm; 61-90cm), IV. soil ecosystem services provided by microorganisms and area plots showing the distribution of the V. bacterial and VI. fungal core phyla. The nine different soil microbiome-associated ecosystem services were estimated based on the relative occurrence of microorganisms known to be involved in 1) improved soil aggregate formation [12], 2) production of antimicrobial agents [13], 3) siderophore production [14, 15], 4) cellulose degradation [16–18], 5) production of antibiotics, [19–21] 6) bioremediation [22, 23], 7) nutrient mobilization [24–27], 8) plant growth stimulation [25, 26, 28–30], and 9) the production of phytohormone-like substances [26] (S2 Table). The metabolic potential of these biological processes was classified into four evenly divided categories: strong, moderate, weak,

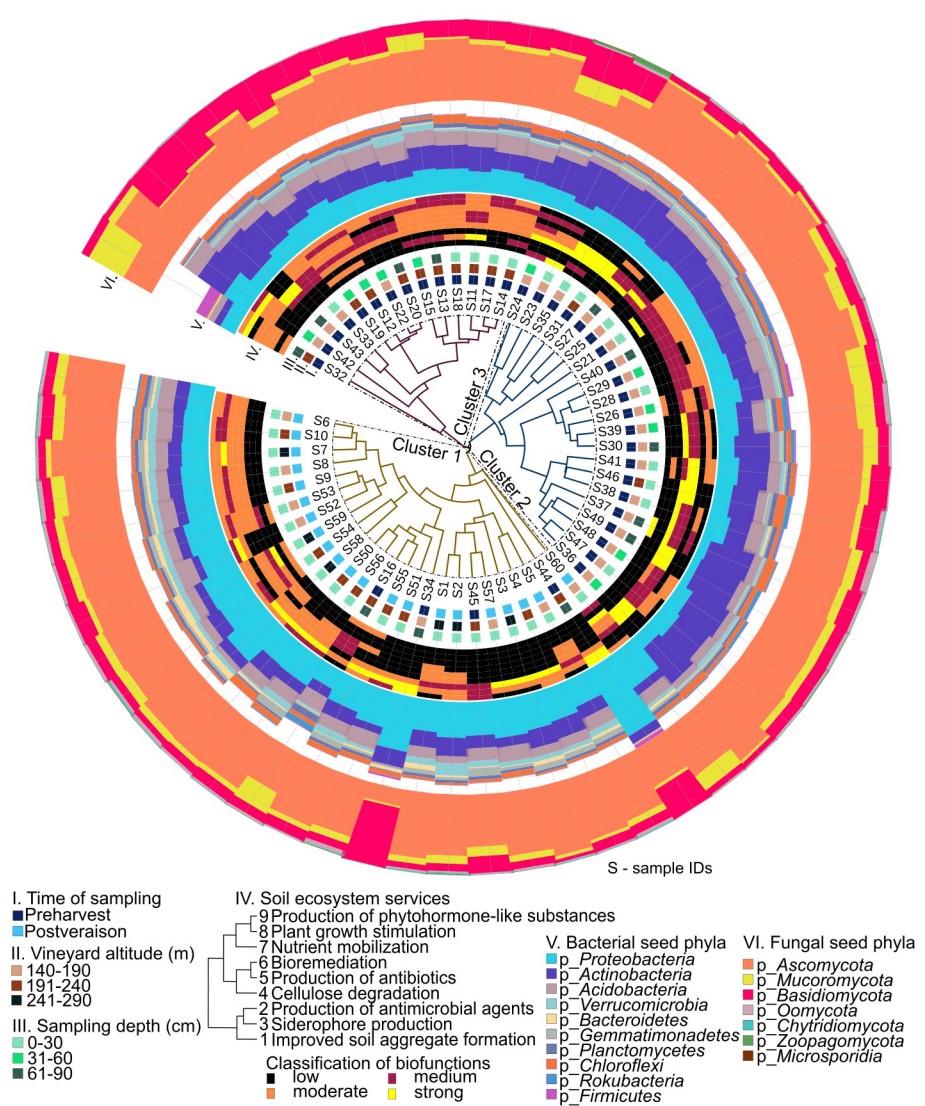

**Fig 5. A detailed look at the 100% core soil microbiota.** Dendrogram illustrating the bacterial and fungal community composition relationships across samples. The tree split into three clusters centred in the middle. The metagenomes associated with distant clusters are marked with different colours. Sample features indicating the I. time of sampling, II. vineyard altitude, III. sampling depth, IV. soil ecosystem services provided by microorganisms and area plots showing the distribution of the V. bacterial and VI. fungal core phyla are indicated with coloured boxes according to the key. Microorganisms play a crucial role in contributing to ecosystem services in the soil. Ecosystem services are classified into categories of "strong," "moderate," "weak," and "low" based on the relative presence of taxonomic groups responsible for specific biological functions. The scaling is determined through referencing scientific literature. The taxons of ecosystem services are a total 100%. If the relative occurrence of an ecosystem service exceeds 75%, it falls under the strong category. If it surpasses 50%, it is considered moderate. If it exceeds 25%, it is classified as weak, and if it is below 25%, it is labelled as low.

and low. Based on the sample taxonomy, the hierarchical clustering analysis resulted in three distinctly different clusters (cluster 1, cluster 2, cluster 3) shown on the circularized taxonomy tree. These clustering patterns did not correlate with the estimated soil ecosystem services. During ripening, the soil fungal microbiota clustered into two groups based on altitude (140–190 m vs. 191–240 m), a pattern not reflected in bacteriomes. Furthermore, parameters such as the time and depth of sampling showed perceivable correlations with soil microbiome

clustering. As such, cluster 1 included soil samples collected postveraison of the Furmint, while cluster 2 and cluster 3 exclusively contained samples collected preharvest. Sixty-two percent of the samples were topsoil samples (0–30 cm), 18% of the samples were middle soil (31–60 cm), and 20% of the samples were from the bottom (61–90 cm). Regarding taxonomical composition varies, at the phylum level, soil core bacterial communities were dominated by *Proteobacteria* (13.30–84.87%), *Actinobacteria* (9.47–63.42%), *Acidobacteria* (0.13–15.99%), *Verrucomicrobia* (0.066%–9.25%), and *Bacteroides* (0.17–9.76%) across all samples. For fungal communities, *Ascomycota* (16.26–90.35%), *Mucormycota* (2.21–50.90%), and *Basidiomycota* (3.40–80.86%) were the dominant phyla.

## The most remarkable taxonomic shifts were observed due to ripening in the soil bacteriome

A phylogenetic analysis technique was used to measure the distances between two sample communities (postveraison vs. preharvest) in terms of the amount of sequence divergence (Fig 6). Samples were spotted on the basis of the Bray-Curtis community matrices. In the case of bacteria, spectacular differences in pattern dynamics were observed in community diversity, resulting in two clusters (postveraison–cluster 1, preharvest–cluster 2) with different spatial ordinations (Fig 6A). Analysis of similarity (ANOSIM) permutation test was used to assess the statistical significance of the separation among groups (p = 0.001). However, overlapping ordination patterns were observed in the case of soil fungal core microbiota during the study period (p = 0.165). Both alpha and beta diversity indices were determined to track marked conversions in the community diversities of soil samples collected at postveraison and preharvest (Fig 6B). Shannon's diversity index was applied to evaluate the species abundance, richness, and evenness of the postveraison and preharvest soil samples. In general, the fungal core microbiota was shown to be more diverse (fungi Shannon's index: 3.63 ± 0.25) than the bacterial core (bacteria Shannon's index: 3.18 ± 0.11). Interestingly, the soil fungal community diversity index did not show significant changes throughout the experimental period; however, a significant decrease in bacteria was observed based on Shannon's index during the growing period (postveraison Shannon: 3.08 ± 0.18 preharvest Shannon: 3.19 ± 0.14, P value < 0.05). The number of observed bacterial and fungal species was also determined, and comparisons were also made between postveraison and preharvest samples (Fig 6C). The proportion of the common species in the two sample groups were shown to be higher for bacteria than for fungi (73.85% vs. 63.79%). Twelve times more common bacterial species (8012 vs. 620 OTUs) were found in our sample cohort. The number of unique species was almost twice as high in the case of *Fungi* than in the case of *Bacteria* (30.35% vs. 16.67%) in preharvest period. Similar but opposite trends were shown in the postverasion samples, where the proportion of the unique bacteria proved to be approximately twice as high as that of fungi (9.48% vs. 5.86%).

## Network analyses elucidate the complex dynamics of the soil fungal interactome

Although microbial biogeography studies focus mainly on the diversity and composition of soil bacteria, global patterns and the impact of biotic interactions on fungal microbial biogeography remain relatively unexplored. To better understand and investigate the complex nature of the microbe–microbe and microbe-community interactions, we performed network analyses on the soil core fungal microbiota (Fig 7). Dissimilarity-based networks were generated to visualize network topology and the changes in microbial relationships. Fungal OTUs with a disproportionately high influence on community members were represented (Fig 7A). We identified 69 influential fungal taxons from most variant OTUs that played an important role

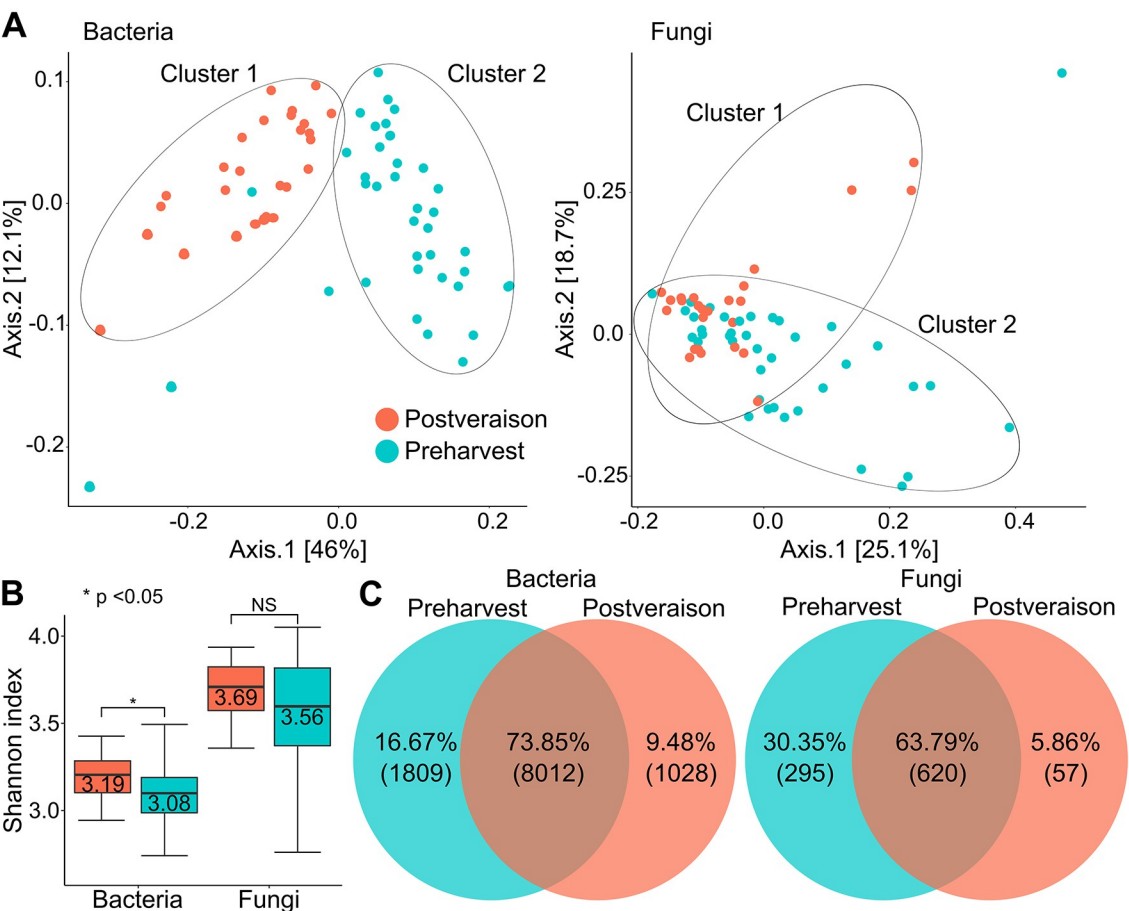

**Fig 6. Ripening had a remarkable impact on the alterations in bacterial communities. A)** PCoA figures summarize the differences between postveraison and preharvest samples according to k_*Bacteria* and k_*Fungi* based on Bray-Curtis distances. For bacteria, distances show two clusters with markedly different taxonomic compositions (p = 0.01). Fungi appeared with overlapping clusters, reflecting more similar taxonomic compositions (p = 0.165). **B)** Group comparison of alpha diversity metrics (Shannon's diversity index). Based on an investigation of the soil core, the bacterial alpha diversity was significantly (P < 0.05) lower in the preharvest samples than in the postveraison samples. **C)** Venn diagram represents the OTU numbers and percentages in k_*Bacteria* (left) and k_*Fungi* (right) comparing preharvest and postveraison samples.

in the community resilience of our sample population. Fungal taxa within these complex eco-logical networks were classified on the basis of their level of connectivity with other taxa. In both of our networks, we found four highly connected taxa (postveraison: o_*Glomerales*, c_*Glomeromycetes*, f_*Glomeraceae*, c_*Agaricomycetes*, preharvest: o_*Mycospnaerellales*, f_*Mycosphaerellacae*, g_*Rhyzopus*, f_*Rhyzopodaceae*) expected to support ecosystem functions. A taxonomy heat tree was also constructed to investigate the significant (P value < 0.05) shifts in the 69 OTUs making up the fungal community architectures (Fig 7B). The tree functions as a key revealing fungal taxonomic lineages of the soil food web. Coloured taxa represent the extents of log2 differences in taxa abundances. Our analysis demonstrated that oligotrophic basidiomycetes such as *Cryptococcus* showed a significant (P value < 0.05) increase with ripen-ing. There is empirical evidence of the importance of rich nodes in regulating the functional potential of soil microbial communities, which enhances the structural robustness of networks against any disturbances. The changes in the degree of the nodes of the 15 most abundant

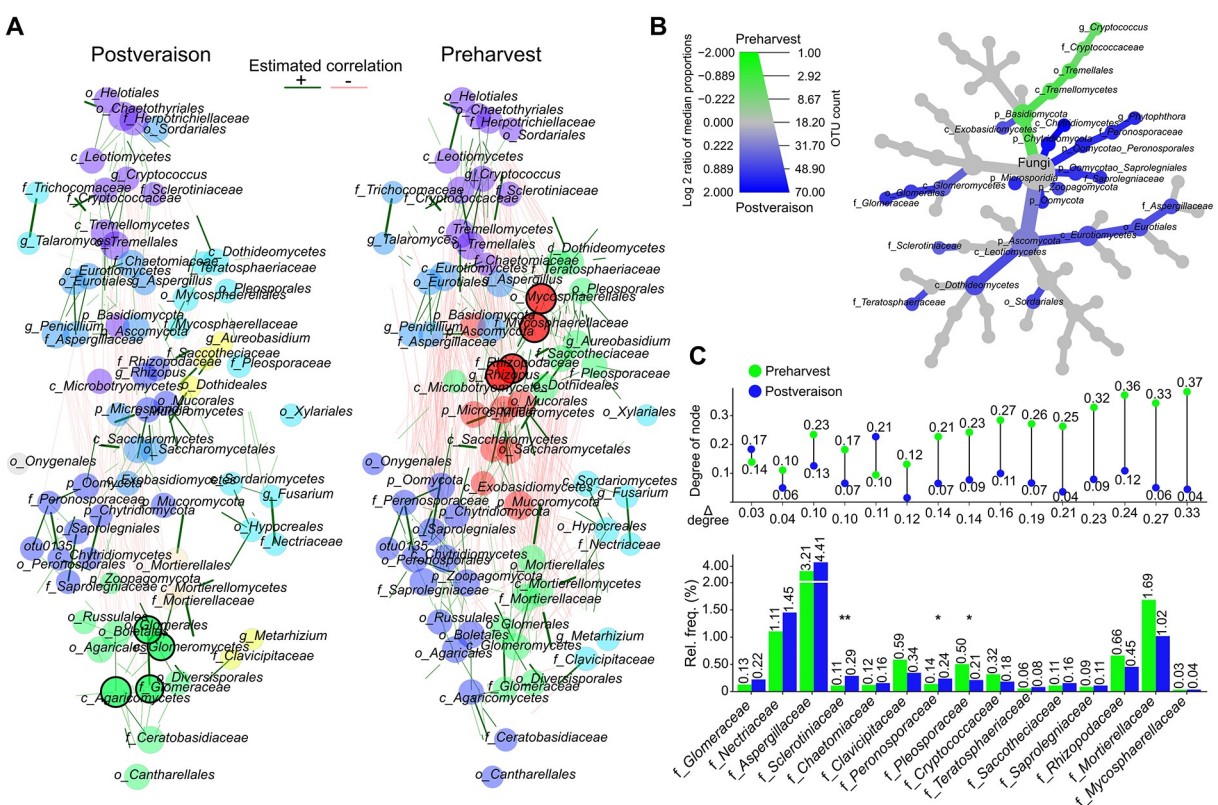

**Fig 7. Network abstractions of the fungal microbial interactome due to ripening. A)** Network analyses were also performed to investigate the symbiotic core fungal core microbiota of the Mád vineyards. The two constructed networks of postveraison (left) and preharvest (right) samples reflect the same architecture. The 69 most common OTUs were selected. Each node represents a core fungal OTU. Data are normalized, green edges correspond to positive correlations and red edges to negative ones. Node colours represent clusters, which are determined using hierarchical clustering. **B)** To examine significant distortions, a community heat tree was constructed to represent the community members of the fungal core showing significant (P value < 0.05) changes in abundances when comparing postveraison vs. preharvest samples. **C)** Representation of the changes in the degree of nodes and relative frequencies of the 15 most abundant fungal families with a strong ability to sustain community resilience. Significant changes in the relative frequency are marked with an asterisk (*) when comparing preharvest and postveraison samples (p < 0.05).

fungal families with high power to sustain community resilience were also analysed (Fig 7C). Family members are arranged in ascending order based on the differences in the degree of nodes. The smallest change was observed in the case of the arbuscular mycorrhizal fungi *Glomeraceae* (Δdegree = 0.03), while the greatest difference was observed in *Rhizopodaceae* (Δdegree = 0.24), *Mortierellaceae* (Δdegree = 0.27) and *Mycosphaerellaceae* (Δdegree = 0.33). Changes in the relative frequencies between the two sample groups are also shown in the figure. Significant (P value < 0.05) changes occurred in the relative frequency of *Sclerotiniaceae*, *Peronosporaceae* and *Pleosporaceae* families when comparing preharvest and postveraison samples.

## Functional redundancy in the 0–90 cm soil microbial community

Possible interconnections between the soil depth-related phylogenetic diversity and functional potential were investigated (Fig 8). In the case of both bacteria and fungi, the topsoil had significantly (P value < 0.05) higher Chao I diversities than the bottom layer (Fig 8A). Functional metagenomics was used to estimate the microbiota-related biological activities in the three

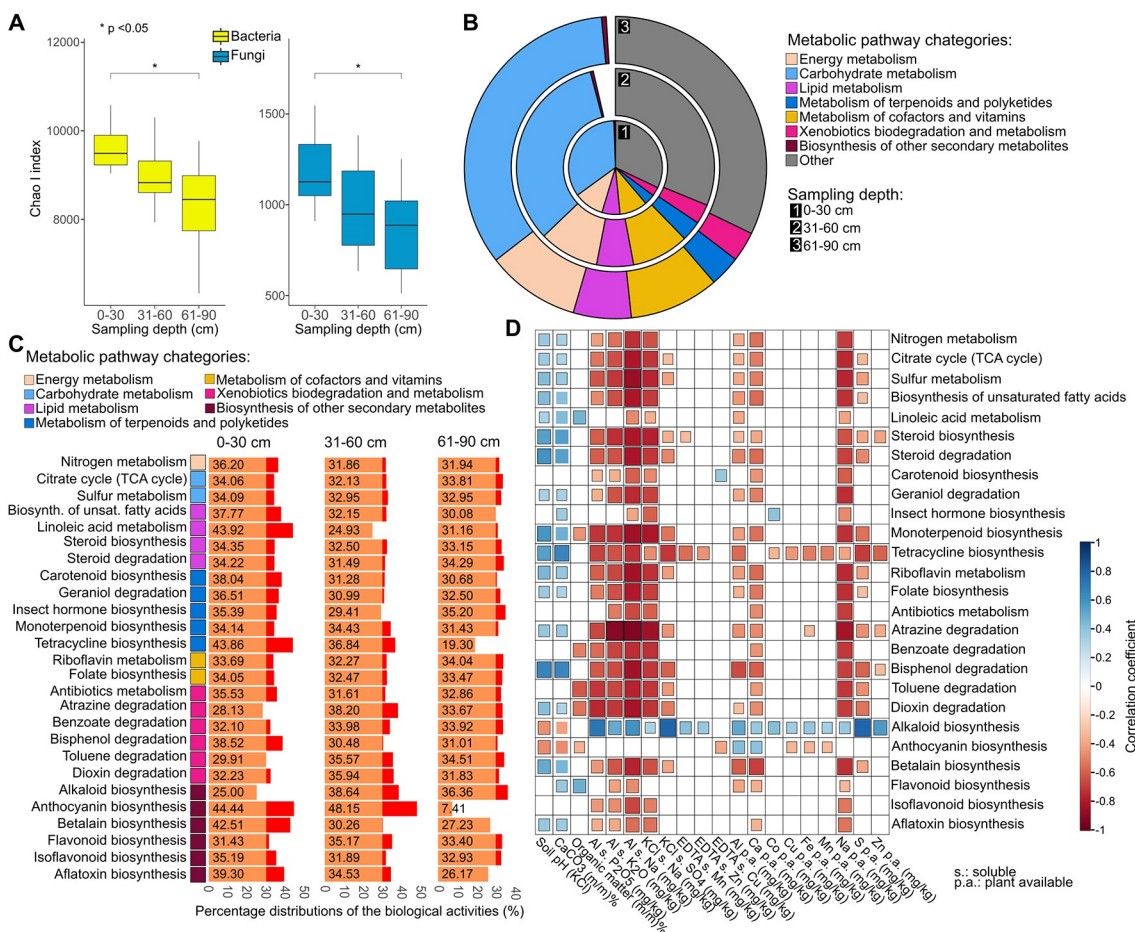

**Fig 8. Microbial diversity and functional attributes across the 0–90 cm soil layers. A)** ChaoI diversity calculations in relation to sampling depth. **B)** KEGG classification of metabolic pathways and the main categories. **C)** Percentage distributions of the biological activities between the 3 sampling depths (0–30 cm, 31–60 cm and 61–90 cm). **D)** Correlation analyses between the soil microbial agrobiochemical cycles and the soil physicochemical properties.

different soil layers (topsoil: 0–30 cm, and in the two subsoil layers: 31–60 cm and 61–90 cm). Regarding the major pathway categories (energy-, carbohydrate-, lipid-, terpenoid and polyketide-, cofactor and vitamin metabolism, degradation of xenobiotics and biosynthesis of secondary metabolites), the trends proved to be shown similar (Fig 8B). The percentage distributions of the 26 detailed biological activities between the 3 sampling depths were also estimated on the basis of the relative abundances of the bacteria encoding enzymes participating in the biological function-related pathways (Fig 8C). Our analyses showed that the vast majority of these biological activities were more pronounced in the topsoil. Tetracycline biosynthesis was very low at 61–90 cm (19.30%) in comparison to that at 0–30 cm (43.86%) and 31–60 cm (36.84%). Soil bacteria are major sources of antibiotics and other compounds that influence interactions among microbial community members and thus impact agrobiogeochemical cycles. Bacterial secondary metabolite biosynthetic potential varied with soil depth. A remarkable difference was shown for bacterial biological functions related to anthocyanin biosynthesis, which was shown to be very weak in the 61–90 cm soil layer compared with the other two soil layers: 46.30% vs. 7.41%. Interestingly, flavonoid biosynthesis was the weakest in

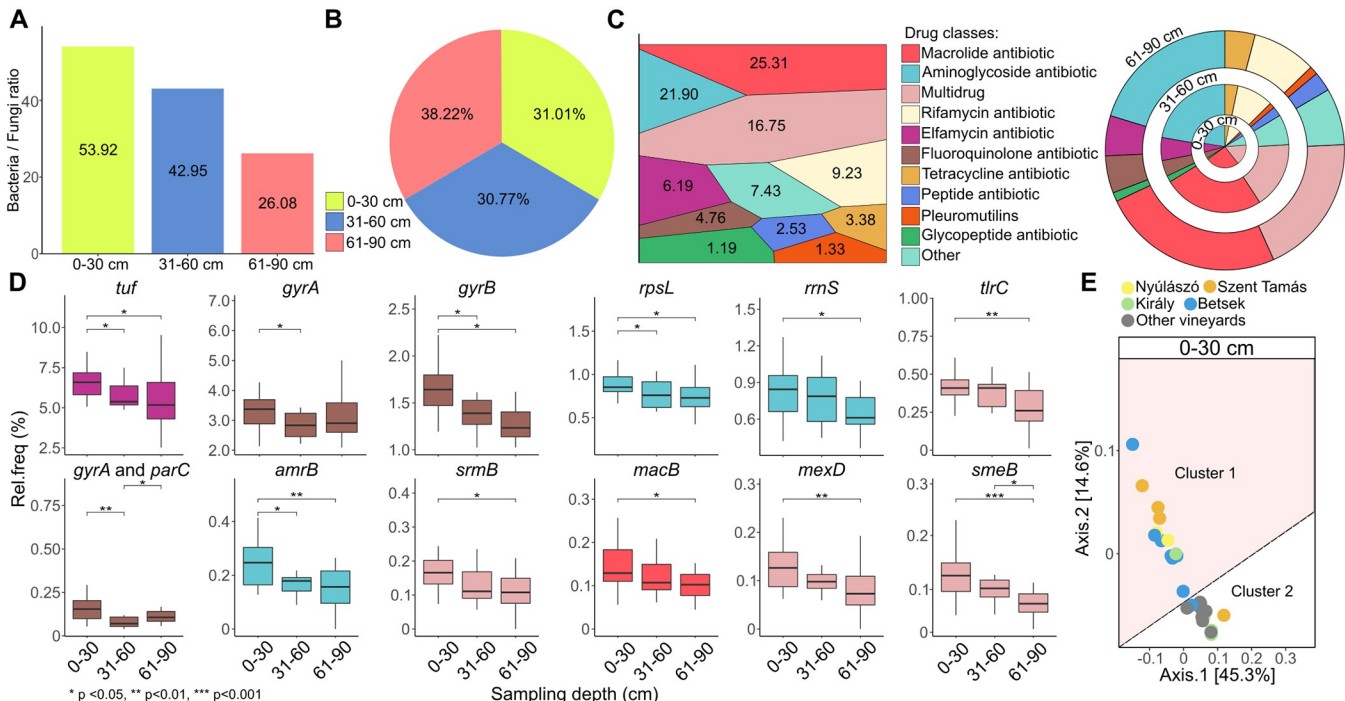

**Fig 9. The soil resistome of Tokaj-Mád vineyards. A)** Bar charts show the Bacterial/Fungal (B/F) ratio and the **B)** occurrence of the ARGs in the different soil layers. Shotgun sequenced data were analysed for taxonomic identification of microbes carrying antibiotic resistance genes using a bioinformatics pipeline. **C)** Distribution of the main antibiotic resistance classes in a Voronoi distribution chart (left) and multiple pie-chart (right) representing the distribution of the main classes across the three different soil layers. **D)** The distribution of the main resistance classes across the three soil layers. **E)** Principal coordinate analysis (PCoA) was used to characterize the distribution of the resistance genes across vineyards in the three soil layers.

the topsoil (31.43%) and the strongest in the 31–60 cm layer (35.17%). Microbiological activities such as nitrogen metabolism (N), carotenoid (Car) and betalain (Bet) biosynthesis was shown to be markedly higher in the topsoil than in the subsoil (N: 36.20% vs. 31.94%, Car: 38.04% vs. 30.68, Bet: 42.51% vs. 27.23%. By investigating the correlation between the 26 biological activities and the soil physicochemical parameters, we found that soil Na (both plant available and nonavailable), similar to the $K_2O$ and $P_2O_5$ contents, stimulated most of the agrobiochemical cycles investigated, except linoleic acid metabolism and alkaloid and anthocyanin biosynthesis (Fig 8D). The pH and $CaCO_3$ content were inversely proportional to the 26 agrobiochemical cycles investigated with the exception that highly calcareous (i.e., alkaline) soils favoured the biosynthesis of alkaloids and isoflavonoids.

## Deciphering of the soil resistome of Tokaj-Mád vineyards

Soil bacterial communities are strongly affected by the bacterial and fungal ratio (B/F). Therefore, we investigated the B/F ratio in all soil layers. The occurrence of *Fungi* was the highest in the bottom layer (61–90 cm) and the lowest in the topsoil (bottom B/F: 26.08, topsoil B/F: 53.92) (Fig 9A). By measuring the relative frequencies in the soil, we found that the highest number of ARGs was present in the bottom layer (61–90 cm: 38.22 vs. mean 0–60 cm: 30.89) (Fig 9B). Vineyard soils are known hotspots for antibiotic resistance genes (ARGs) and potential pathogens; therefore, in this study, the ARG profiles of the soil microbiota of the Mád wine region were also thoroughly investigated. Regarding the main antibiotic resistance classes, macrolide antibiotic resistance was the most abundant (25.31%), followed by aminoglycoside

resistance (21.90%) (Fig 9C). The occurrence of multidrug resistance was also notable (16.75%). The distribution of the main resistance classes across the three soil layers showed similar profiles (Fig 9C). Based on the occurrence of the most abundant resistance genes (rel. freq. > 0.05%), in general, their accumulation was significantly (p < 0.05) higher in the topsoil. In the case of resistance genes such as *tuf* (elfamycin), *gyrB* (fluoroquinoione), *rpsL* and *amrB* (aminoglycoside), the occurrence of resistance factors was the highest in the 0–30 cm soil layer (Fig 9D). Interestingly, considering the accumulation of the multidrug resistance factors *tlrC*, *srmB*, *mexD* and *smeB*, such as *rrnS* (aminoglycoside) and *macB* (macrolide) in the topsoil, the levels were significantly (P value < 0.05) higher only in relation to the lowest layer (61–90 cm). For the second most frequent resistance factor, *gyrA* (fluoroquinolone), the lowest levels were measured in the 31–60 cm layer. Principal coordinate analyses (PCoA) were also used to characterize the distribution of the carried resistance genes of the four historical vineyards (Betsek, Szent Tamás, Király, Nyúlászó) and the other (Fig 9E). In the case of the topsoil samples, different trends were shown when comparisons were made between the samples derived from the four historic vineyards (cluster 1) and those from the 6 surrounding (Ősz-hegy, Sarkad, Hold-völgy, Úrágya, Danczka, and Szilvás) areas (cluster 2).

## Discussion

Viticulture is a significant economic sector globally (A common framework for sustainability indicators in the wine sector: Dream or reality?, Precision viticulture: The state of the art). However, climate change and the current economic situation pose new challenges [31]. To overcome these threats, a better understanding of the soil microbiota is essential. The Tokaj wine region boasts a unique microclimate that encourages grapes to develop botrytis, a type of fungus that results in naturally sweet noble wines. This fungus enriches the wine with aromas similar to the fragrance of certain trees such as linden and robinia. Additionally, it adds flavors of sundried apricot and quince, which are considered the hallmarks of Tokaj wines. The natural conditions of the region, such as its altitude, bedrock, micro and meso-climatic factors, and the concentration of trace elements in the soil, create the perfect environment for producing single-vineyard wines. These factors combine to produce wines with unique characteristics and exceptional quality, especially in the Mád wine region. Wine terroir characteristics are related to region-specific microbial community compositions [6, 32]. The vineyard soil is the primary source of bacteria, yeasts, and microscopic fungi living and growing on grape surfaces, which are important for winemaking root function [33].

Advances in high-throughput NGS technologies have led to a new era of microbiome research function [34]. However, due to the unique characteristics of the existing statistical and experimental methods of these surveys, ecological interaction networks remain a considerable challenge. In this study, mutual interactions between soil fungal and bacterial diversity, functional metagenomic potential and soil chemistry were investigated [35].

Concerning the taxonomic composition of vineyard soils, it can be estimated that microbial fingerprints of the historic vineyards are distinguishable from those of other areas in the Tokaj-Mád wine region, a distinction that arises from the exhibited taxonomic similarities. There was a positive correlation between soil bacterial diversity and the vast majority of the examined soil physicochemical characteristics, while fungal diversity showed a negative correlation with these characteristics. Manganese is an essential micronutrient for plant physiology and growth. As an activator and cofactor of various metalloenzymes and a catalyser of redox reactions, Mn is involved in the structure of photosynthetic proteins and enzymes; thus, its deficiency causes oxidative stress and negatively affects root function [36]. The mild positive correlation (mean r: 0.35) observed between the soil bacterial diversity and soil manganese

content might be credited to the presence of plant-available Mn-mobilizing bacteria such as plant growth-promoting *Pseudomonas* spp. (rel. freq.: 0.32%), *Rhizobium* spp. (rel. freq.: 0.075%), *Bacillus* spp. (rel. freq.: 0.07%) [36]. There was a mild negative correlation (r: -0.48) between fungal Simpson diversity and soil organic matter. This is possibly because *Fungi* are 2–3 orders of magnitude less abundant than Bacteria (rel. freq. *Bacteria*: 99.83% vs. rel. freq. *Fungi*: 0.17%). Regardless, we also found that the diversity of Fungi in the soil was moderately higher (Shannon: 3.59±0.27, Simpson: 0.93±0.016) than that of *Bacteria* (Shannon: 3.08±0.17, Simpson: 0.85±0.017). In the case of Mádi vineyards, diversity values were found to lag behind those measured in other studies. Specifically, when compared to an Argentinian study conducted in a recently established Malbec wine region, where Shannon diversity was reported between 7–8 for bacteria and 6–8 for fungi [37]. Similarly, a study investigating the soil microbiota in the Italian province of Trentino found Bacterial Shannon diversity between 6–7 and Fungal Shannon diversity between 4–6 [38] Furthermore, a German study examining the impact of soil management practices on soil microbial diversity in nine German vineyards reported Bacterial Shannon diversity ranging from 6.6–7.2 and Fungal diversity from 3.2–5 [39]. Moreover, in our case, fungal diversities were found to surpass the diversity values of bacteria.

The functional stability of the microbial communities depends on their compositional plasticity. Ecosystems are highly susceptible to cumulative stressors, both biotic and abiotic [40]. It is crucial to have a better understanding of the relatively conserved core microbiota. Therefore, we characterized the 100% core soil microbiota of the Tokaj-Mád vineyards. We explored the influence of variables including altitude, time, and sampling depth on the composition of the soil microbiomes, along with their repercussions on ecosystem services.

By comparing the beta diversities of the samples collected postveraison and preharvest, we investigated the temporal dynamics of the 100% soil core microbiota of Tokaj-Mád [2]. Our investigation revealed significant changes in the soil bacterial core, particularly in the two main phyla, *Proteobacteria* and *Actinobacteria*, attributed to the ripening process. The changes in the fungal core microbiome were ~~not~~ less pronounced in comparison to the bacteriome. In the case of the balanced microbiota of the Mád vineyards, there were no significant differences in the community alpha diversities. In the future, a better understanding of which factors influence these bacteria may provide insights into management practices to shape and craft individual wine properties.

We observed that temporal factors exerted a marked influence on the soil bacterial community composition. The observation that diverse soil fungal symbiomes are more resistant to external interventions is confirmed by the fact that fungal compositions showed a higher correspondence due to ripening, indicating that community resilience increases with diversity [41]. Computational tools allowed us to perform network comparisons of dynamic systems. The functionality of complex networks relies on their structural robustness. Their resilience, and therefore their ability to maintain symbiotic states, depends on how quickly they are able to respond to all kinds of external environmental changes and internal disturbances [42]. The cohesive forces of the fungal core interactomes were investigated on the basis of the estimated positive and negative correlations.

In Mád, the climatic conditions in August and September are relatively dry, which helps the grapes to achieve full biological ripeness before they start to overripen or shrivel. We have discovered that microbial interactions are more active during the ripening period, which leads to enhanced community modularities in postveraison and preharvest samples. A strong community safety net is represented by networks with dense connections between the nodes. The more connections a microbe has, the more important its role in the whole community. Our research focused on identifying the most strongly anchored fungi which play a crucial role in

maintaining the community safety net in equational communities. This information is of great importance because removing these hubs can increase the fragility of the system. *Glomeraceae* was found to play a similarly prominent role in maintaining community resilience. This is not surprising regarding the fact that the family increases grapevine growth and nutrition by better access to soil nutrients and by activating the regulation of plant transport proteins for phosphorus (P), nitrogen (N), and other elements. *Rhizopodaceae* and *Mycosphaerellaceae* have been found to be more influential throughout the ripening period. Furthermore, members of *Rhizopus oligosporus* and *Rhizopus oryzae* are superior in the saccharification of starches.

Soils with high pH and rich in calcium are reported to be optimal for grapevines, making nutrients more available and easily digestible. High soil pH and carbonate expose plants to abiotic stress that enhances their alkaloid concentration. In addition to stress management, these mutualistic relationships also shield the host plant against competitors or predators [43].

Understanding the relevance of soil microbial communities in soil agro-biochemical cycles is crucial. Similar to $K_2O$ and $P_2O_5$, soil Na, encompassing both plant-available and non-available forms, demonstrated correlations with most of the examined microbial agro-biogeochemical cycles, with the exception of linoleic acid metabolism, alkaloid biosynthesis, and anthocyanin biosynthesis. The soil pH and $CaCO_3$ contents exhibit an inverse relationship with most of the 26 biological functions investigated, except for the fact that highly calcareous soils (i.e., alkaline soils) promote the biosynthesis of alkaloids and anthocyanins. Extensive anthropogenic activities accelerate the dissemination of antibiotic resistance genes globally. Being a huge reservoir, soil facilitates the exchange of ARGs among potential pathogenic bacteria, impacting human health through various transmission mechanisms [8].

The impact of soil on antibiotic resistance genes (ARGs) is a global concern, but we have limited knowledge about the ARG profiles of vineyard microbiota. The unique accumulation of resistance gene profiles found in the topsoil samples of Betsek, Király, Nyúlászó, and especially Szent Tamás, could be attributed to changes in viticulture practices. Historic vineyards typically follow traditional winemaking methods that have been passed down for generations. This preserves the authenticity and heritage of winemaking. Research has shown that ARGs, or antibiotic-resistant genes, are more abundant in the topsoil when fungal biomass is higher. This suggests that fungi secrete antimicrobial compounds to outcompete bacteria, which creates a selective pressure for antibiotic-resistant bacteria to emerge [44]. This theory was also supported by our data. The highest proportion of the ARGs was found in the bottom soil, where the fungal ratio was the highest.

## Conclusion

Mád is part of the oldest wine regions in the world, where Furmint is the favoured varietal. One of our major aims was to characterize the microbial terroir of the UNESCO Tokaj-Hegyalja Mád region. The most dominant and discriminant bacterial and fungal taxa were well characterized. Fungal community diversity was markedly higher than that of Bacteria, suggesting a more resilient fungal community network. Arbuscular mycorrhizal fungi were shown to have important functions in the dynamic assembly and stabilization of the core communities of the Mád viticultural zones. There are differences between the most famous historical crusades and those located nearby. Differences represent a nonrandom distribution in the soil microbiota that is associated with topographic, geologic, climatic, biotic and abiotic factors shaping the microbiota. By elaborating the interplay between soil physicochemical parameters and microbiome-related agrobiochemical cycles, this study reveals possible applications for wine fermentation management, with the opportunity to develop predictive models on the basis of community responses to different conditions.

## Materials and methods

### Sampling

Soil sampling was carried out by the Centre for Precision Farming R&D Services (University of Debrecen). Collection of the soil samples from the Tokaj-Mád wine growing region was permitted by the Tokaj-Hegyalja mountain commune. Mád lies between 21˚16′33.52 east longitude and 48˚11′32.08 north latitude. The grape cultivation areas from Mád are part of Tokaj Wine Region. Mád is located in a valley, which is surrounded by forested meadows from the north and mountain ranges planted with vines from the east and west. Soil samples were collected in Mád, Hungary, in the Ősz-hegy, Sarkad, Hold-völgy, Úrágya, Danczka, Nyúlászó, Szent Tamás, Király, Betsek and Szilvás vineyards on 10 August (postveraison–only soil) and 13 September (preharvest–soil) 2021. Samples were transported to the Faculty of Agricultural and Food Sciences and Environmental Sciences, Center for Complex Systems and Microbiome Innovations (University of Debrecen) and were stored at −80 ˚C until further analysis. Soil samples were obtained at pre-defined management plots representing typical relief, slope exposure and vineyard quality classes. Sampling has been carried out with a P4000 type truck-mounted soil sampler platform (Veris Technologies (Salina, KS, USA). Soil cores were taken using disposable PETG liners as sampler tubes; the cutting shoe was cleaned with 3% hydrogen-peroxide solution following every sampling point. Each sample, consisting of 15–20 sub-samples, was taken from two rows of each zone, between vine plants. The depth of coring varied depending on the presence of coarse textures, such as stones and rocks in the soil profile, and the depth of bedrock, resulting in different sampling depths and, therefore, different number of samples from the sampling locations. Altogether 60 samples were collected from 0–30 cm, 31–60 cm, 61–90 cm depth zones. Metagenomic DNA was isolated from each soil sample pool. Out of the total 60 samples, 37 (~62%) were taken from the topsoil (0–30 cm), 11 samples (~18%) from the middle soil layer (31–60 cm), and 12 samples (20%) from the bottom layer (61–90 cm). The prevalence of topsoil samples is attributed to the challenges associated with sampling at greater depths, given the presence of hard rock and a thin, fertile soil layer. Total elemental analysis of soil was performed at Central Laboratory of Agricultural and Food Products, University of Debrecen all analysis methods were according to standards [45]. The determination of major, minor and trace elements was performed via ICP-OES (Al.s. $P_2O_5$, $K_2O$, Na; KCl.s. Mg, $SO_4$-S; EDTA.s. Mn, Zn, Cu; p.a. Al, Ca, Co, Cu, Fe, Mn, Na, S, Zn), potenciometry (pH), gas volumetry ($CaCO_3$) and colorimetry (OM). Out of the 60 soil samples collected, 42 samples were successfully evaluated through total elemental analysis.

### Soil sample processing

Soil samples (10 g) were weighed and placed into a 50 mL centrifuge tube to which 20 mL of phosphate buffered saline (PBS) (Biosera, France) was added. The tubes were then shaken at room temperature for 30 minutes at 150 rpm in New Brunswick Benchtop Incubator Shaker Innova 40/40R (Eppendorf, Germany). The supernatant was then transferred to a new 50 ml tube, and the washing step was repeated. This was followed by transferring the supernatant into a 50 mL tube with centrifugation at $200 \times g$ for 1 minute. The supernatant was transferred into a new 50 ml tube and centrifuged at $12,600 \times g$ for 25 minutes. The pellet was suspended in 3 ml of sterile PBS and stored at -20˚C.

### Nucleic acid extraction and shotgun sequencing

DNA extraction from soil samples was performed using the DNeasy® PowerSoil® Pro Kit (Qiagen, Germany) following the manufacturer's instructions. Minor modifications were

made to optimize the DNA extraction. In brief, 750 μl of soil sample supernatant was centrifuged at 21,000 × g for 5 minutes. The pellet was dissolved in Solution CD1, transferred into PowerBead Pro Tubes (Qiagen, Germany) and incubated at 65°C for 10 minutes. With the use of a MagNA Lyser Instrument (Roche Applied Sciences, Germany), samples were lysed twice at 6,000 × RPM for 30 seconds each. Finally, 70 μl of Solution C6 was added and incubated at room temperature for 5 min before centrifugation. DNA concentrations were determined fluorometrically using a Qubit® Fluorometric Quantitation HS dsDNA Assay kit on a Qubit® 4.0 Fluorometer (Thermo Fisher Scientific, USA). The DNA was fragmented and 5' Adapter:5'–

`AGATCGGAAGAGCGTCGTGTAGGGAAAGAGTGTAGATCTCGGTGGTCGCCGTATCATT–3'`,
and 3' Adapter: 5'–

`GATCGGAAGAGCACACGTCTGAACTCCAGTCACGGATGACTATCTCGTATGCCGTCTTCTG`
`CTTG–3'` adapters were used. Shotgun sequencing was conducted on an Illumina NovaSeq 6000 instrument (Illumina, USA) with a 150-bp paired-end sequencing run at Novogene Co. Ltd, China. The sequencing yielded a minimum of 20 million reads per sample. To ensure the availability of 20 million reads per sample for shotgun sequencing, we achieved this by re-isolating from each sample until obtaining the required purity OD260/280 = 1.8–2.0 and concentration $\geq$ 10 ng/μL for each metagenomic isolate.

## Bioinformatics analyses

Shotgun sequencing data were analysed with the SqueezeMeta program, a fully automated metagenomics pipeline on default settings without trimming. In SqueezeMeta, the sqm_reads.pl script was used to perform taxonomic and functional assignments on individual reads rather than contigs [46]. Every sample was analysed individually. To run the program efficiently, we used the KIFÜ Hungarian High Performance Computing Competence Center (HPC CC). For every sample, 48 CPU cores and 130 GB RAM were selected. The program uses GenBank, a comprehensive database that contains publicly available nucleotide sequences of species for taxonomic profiling [47]. Ecosystem services are classified into categories of "strong," "moderate," "weak," and "low" based on the relative presence of taxonomic groups responsible for specific biological functions. The scaling is determined through referencing scientific literature. If the relative occurrence of a taxon exceeds 75%, it falls under the "strong" category. If it surpasses 50%, it is considered "moderate". If it exceeds 25%, it is classified as "weak", and if it is below 25%, it is labelled as "low". For functional analysis, the eggnog [48] and KEGG databases were used [49]. The KEGG program uses DIAMOND as a sequence aligner [50]. Resistome analysis was performed with the RGI Resistance Gene Identifier program [51] with the kneaddata program (https://huttenhower.sph.harvard.edu/kneaddata/) for data preprocessing. The resulting data were downloaded from the HPC server. For further data processing, the phyloseq R package was used with different accessory programs. [52]

## Data visualization

The results were presented via principal coordinates analysis (PCoA). Figures were visualized with the ggplot2 R package [53]. The correlations between two components were calculated to determine important internal relationships, graphically shown through a heatmap generated by the corrplot package in R [54]. For network analysis, the NetCoMi R package was used [55]. The abundance of each taxon was plotted using the heat_tree function of the R package metacoder, excluding low-abundance taxa [56]. To decipher the relationships between bacterial families and soil characteristics, canonical correspondence analysis (CCA) was performed using the "vegan" package in R 3.3.2 software [57]. Dendrogram visualizations were generated

using the dendextend package [58]. To acquire differential taxa, LefSe analysis and intergroup rank-sum test analysis were conducted. LefSe was conducted by means of the nonparametric Kruskal–Wallis test, which used linear discriminant analysis (LDA) to estimate the influence of the abundance of each component (taxon) on the difference effect, thus illustrating the communities or taxa with significant differences in effect on sample division [59].

## Statistical analysis

To analyse the significant differences among more than two groups, the nonparametric Kruskal–Wallis test of the stats package in R was used with the general threshold of a *P* value of ≤ 0.05. For principal coordinate analysis (PCoA) statistical analysis ANOSIM permutation test was used to assess the statistical significance of the separation among groups [60] using the R "vegan" package [57].

## Supporting information

**S1 Fig. A hierarchically clustered heatmap was constructed on the basis of the organic and mineral contents of the 42 bulk soil samples of the Mád vineyards.** There were six distinct clusters identified according to the soil physicochemical properties (SPP): Cluster SPP1-Cluster SPP6.
(PDF)

**S1 Table. Organic and mineral content analysis was conducted on 42 soil samples collected from vineyards in Mád.** The origin of each sample was specified, indicating the vineyard plot and soil depth.
(DOCX)

**S2 Table. Soil ecosystem services provided by microorganisms.** The nine different soil microbiome-associated ecosystem services were estimated based on the relative occurrence of microorganisms known to be involved in 1) improved soil aggregate formation, 2) production of antimicrobial agents, 3) siderophore production, 4) cellulose degradation, 5) production of antibiotics, 6) bioremediation, 7) nutrient mobilization, 8) plant growth stimulation, and 9) the production of phytohormone-like substances. The metabolic potential of these biological processes was classified into four evenly divided categories: 4-strong, 3-moderate, 2-weak, and 1-low.
(DOCX)

## Acknowledgments

We acknowledge KIFÜ for awarding us access to resource based in Hungary. This project has been implemented with the support provided by the Ministry of Innovation and Technology of Hungary from the National Research, Development and Innovation Fund, financed under the TKP2021-NKTA funding scheme.

## Author Contributions

**Conceptualization:** Judit Remenyik, László Csige, István Szepsy Jnr, László Stündl, Attila Csaba Dobos, Melinda Paholcsek.

**Data curation:** Péter Fauszt, Anna Anita Szilágyi-Rácz, Gábor Fidler, Melinda Paholcsek.

**Formal analysis:** Péter Fauszt.

**Funding acquisition:** László Csige.

**Investigation:** Judit Remenyik, Péter Dávid, Anna Anita Szilágyi-Rácz, Erzsébet Szőllősi, Zsófia Réka Bacsó, Krisztina Molnár, Csaba Rácz, Zoltán Kállai, László Stündl, Attila Csaba Dobos, Melinda Paholcsek.

**Methodology:** Judit Remenyik, Péter Dávid, Anna Anita Szilágyi-Rácz, Attila Csaba Dobos, Melinda Paholcsek.

**Project administration:** István Szepsy Jnr, Zoltán Kállai.

**Resources:** László Stündl.

**Supervision:** Judit Remenyik, László Csige, Attila Csaba Dobos, Melinda Paholcsek.

**Validation:** Melinda Paholcsek.

**Visualization:** Péter Fauszt, Anna Anita Szilágyi-Rácz, Gábor Fidler.

**Writing – original draft:** Melinda Paholcsek.

**Writing – review & editing:** Melinda Paholcsek.

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
