## [Decision Letter · Decision Letter 0]

22 Nov 2023

PONE-D-23-21640Exploring the Interplay Between the Core Microbiota, Physicochemical Factors, Agrobiochemical Cycles and Resistome in the Soil of the Historic Tokaj Mád Wine RegionPLOS ONE

Dear Dr. Paholcsek,

Thank you for submitting your manuscript to PLOS ONE. After careful consideration, we feel that it has merit but does not fully meet PLOS ONE’s publication criteria as it currently stands. Therefore, we invite you to submit a revised version of the manuscript that addresses the points raised during the review process. Please carefully consider the reviews provided and address the major revisions that are needed. A number of reviewers discussed the need for greater focus in the manuscript and clearly defined questions that could be directly addressed by the data provided. In the current form there are too many questions, and some questions with unclear support from the data provided. A number of reviews mentioned that sections need to be more clearly aligned with the data provided in the figures. There are detailed suggestions for improvements and increased clarity. Make sure to address the suggestions and comments of all reviewers. 

We look forward to receiving your revised manuscript.

Kind regards,

Theodore Raymond Muth

Academic Editor

PLOS ONE

Reviewers' comments:

Reviewer's Responses to Questions

**Comments to the Author**

1. Is the manuscript technically sound, and do the data support the conclusions?

Reviewer #1: Partly

Reviewer #2: Yes

Reviewer #3: Partly

Reviewer #4: Partly

Reviewer #5: Yes

2. Has the statistical analysis been performed appropriately and rigorously? 

Reviewer #1: Yes

Reviewer #2: Yes

Reviewer #3: I Don't Know

Reviewer #4: Yes

Reviewer #5: I Don't Know

3. Have the authors made all data underlying the findings in their manuscript fully available?

Reviewer #1: Yes

Reviewer #2: Yes

Reviewer #3: No

Reviewer #4: Yes

Reviewer #5: No

4. Is the manuscript presented in an intelligible fashion and written in standard English?

Reviewer #1: Yes

Reviewer #2: Yes

Reviewer #3: Yes

Reviewer #4: Yes

Reviewer #5: Yes

5. Review Comments to the Author

Reviewer #1: REVIEW OF MANUSCRIPT (PONE-D-23-21640) ENTITLED „EXPLORING THE INTERPLAY BETWEEN THE CORE MICROBIOTA, PHYSICOCHEMICAL FACTORS, AGROBIOCHEMICAL CYCLES AND RESISTOME IN THE SOIL OF THE HISTORIC TOKAJ MÁD WINE REGION” BY REMENYIK J. ET AL.

The soil microbial community plays a pivotal role in the growth and vigor of cultivated plant species, so the research aim is sensible and well-articulated. Soil samples, originated from ten vineyards of the Tokaj-Mád wine region were analyzed and compared based on a metagenomics research strategy. Characteristics of the soil microbial community were evaluated at three depths and at two-time points. The first question to come to light is related to the sampling strategy; authors indicated that altogether 60 soil samples were collected, nevertheless one sample consisted of 15-20 subsamples: 1) How the subsamples were pooled, soil or community DNA were combined? 2) Why the percentages of top, middle and deep soil samples differ? 3) What is the reason of soil sample number discrepancy mentioned in the text and S1 Fig. and Table (60 vs 42 respectively)?

In order to correctly investigate the qualitative diversity and the community structures of the samples, rarefaction and/or rank abundance curves generation is critical. 4) Please demonstrate/ validate that the number of OTUs are saturated in all group samples to get enough sequencing depth. 5) By the way, for some reason almost all of the figures and their labels are not sharp in the pdf version of the manuscript, however downloaded tiff files were legible. 6) Please pay more attention to the proper preparation of the figure and table legends; do not use incorrect, inconsistent or unexplained abbreviations in the figures, or such that are not common knowledge.

A few comments about the applied techniques and their presentation. The performed methods and techniques meet the current technological standards and are suitable for answering the scientific questions formulated in the objectives. Shotgun metagenomics is a powerful approach

to assess microbial composition, diversity, and functional potential, 7) however, it is hard to believe that “direct analysis of individual reads rather than contigs”, would provide more reliable results, especially in the case of short reads (2 × 151 bp). 8) As for the quantity of reads generated from the samples – 20 million reads per sample – it highly questionable that such a uniform high-quality read count could be correct. 9) Although it's a small thing, MagNA Lyser Instrument indicates oscillation frequency (speed) in revolutions per minute (rpm) not in G-force. Nevertheless, these are minor remarks, not affecting unfavorably the value of the work presented. The results and discussion section nicely follows the logic of the materials and methods. It mostly presents the findings clearly. In general, the writing style of the manuscript is proper, and the text contains only a few typographic errors and ambiguous phrasing, however personally I would like to see a much more intense and detailed result comparison with previous knowledge during the discussion.

Reviewer #2: Reviewer Comment for Manuscript PONE-D-23-21640: Exploring the Interplay Between the Core Microbiota, Physicochemical Factors, Agrobiochemical Cycles and Resistome in the Soil of the Historic Tokaj Mád Wine Region

Summary of the studies:

Shotgun metabarcoding was used in the assessment of microbial-terroir in the Tokaj-Mád vineyards of Hungary. The mutual interactions between soil diversity, functional potential and soil chemistry were investigated across the 0 - 90 cm soil layers (0 - 30 cm, 31 - 60 cm and 61 - 90 cm). The results from 60 samples indicated that fungal phyla, Ascomycota, Basidiomycota, and Mucoromycota were operating at 97% operational taxonomic units. By investigating the correlation between the 26 biological activities and the soil physico-chemical parameters, the authors found that soil Na (both plant available and non-available), similar to K2O and P2O5 contents, stimulated most of the agrobiochemical cycles studied, except linoleic acid metabolism, alkaloid and anthocyanin biosynthesis. Soil bacterial diversity was positively correlated with the vast majority of the examined soil physico-chemical characteristics, while fungal diversity was negatively correlated with these characteristics. The pH and CaCO3 content were inversely proportional to the 26 agrobiochemical cycles investigated with the exception that highly calcareous (i.e., alkaline) soils favoured the biosynthesis of alkaloids and isoflavonoids. From the discussion, the natural conditions, altitude, bedrock, micro- and mezzo-climatic factors, and concentration of trace elements in the soil render the Mád wine region suitable for making single-vineyard wines, thus contributing to some of their distinctive characteristics and superb quality. The authors observed that temporal factors exerted a marked influence on soil bacterial community composition. Fungal compositions showed a higher correspondence due to ripening confirms the general observation that community resilience increases with diversity. This also suggests that diverse soil fungal symbiomes are more resistant to external interventions. The authors conclude that, understanding the contribution of soil microbial communities to soil agrobiochemical cycles is critical for predicting terrestrial ecosystem feedbacks.

General Remarks:

This is an interesting and well written manuscript. The title is appropriate for the content of this article. The abstract is concise and accurately summarizes the essential information of the paper. The introduction provides an excellent, generalized background of the topic and this gives the reader an appreciation of the wide range of applications to the technology (nucleic acid extraction and shotgun sequencing) used for this research. The authors collected a unique dataset and the data were analyzed following laid down scientific methodology. The experimental design is standard and appropriate for this study. The authors provided adequate information especially in the materials and methods to make this research experiment capable of been reproduced by a capable research scientist.

The main strength of this paper is that it addresses an interesting topic and it contributes to our understanding on the importance of soil resistome studies and the associated antibiotic resistance genes (ARG) dissemination mechanisms. The manuscript also finds a novel solution based on a carefully selected set of rules and procedures and finally provides a clear solution to the research questions asked. Literature cited in the manuscript, are relevant for this study. The figures and tables are well organized, easy to interpret and follows a systematic and a logical order. It improves our understanding of the importance of the spatial variability of soil resistomes and how ARG depends on microbial community structure as well as its functional potential.

Generally, the study is of great significance and contributes extensively to our understanding of soil microbial communities to soil agrobiochemical cycles is critical for predicting terrestrial ecosystem feedbacks in the study area. I recommend this manuscript (PONE-D-23-21640) for publication after some corrections have been taken into account. Thus, to make the manuscript more reader friendly to an international audience, I suggest that the authors, if possible, should rephrase the title of the manuscript, some sentences in the manuscript and also include the significant level (e.g., 0.05, 0.01, 0.001) and the p - values (e.g., p < 0.03), if possible, into their findings where significant or not significant and/or where positive or negative corelation (r = 0.00; p -value) is reported in the manuscript.

Specific Remarks:

Pg. 3

Line 83 - 84: Metabolites produced by the rhizobiome are important contributors to the quality, fragrance and distinctive personality of the products (5,6).

Please replace “personality” with “quality”.

Pg. 4

Line 86 - 88: The adaptation abilities of the microbiota depend on community plasticity. Network analysis-based approaches can help in better understanding the intricate nature of microbe‒microbe and microbe-community interactions.

Please rephrase the above sentence as: The adaptation abilities of the microbiota depend on community plasticity. Network analysis-based approaches can help understand the intricate nature of microbe‒microbe and microbe-community interactions.

Pg. 10

Line 244 - 245: At the phylum level, 90.0% of the Bacteria and 70.0% of the Fungi OTUs were present in our sample population.

Please edit as: “At the phylum level, 70% and 90% of Fungi OTUs and Bacteria were present in our sample population, respectively”.

Pg. 11

Line 255 - 256: Comprehensive comparisons of the soil samples of the Tokaj-Mád vineyards were made based on the hierarchical clustering of the bacterial and fungal 100% core microbiota (Fig. 5).

Please rephrase the above sentence as: Comprehensive comparisons of the soil samples of the Tokaj-Mád vineyards were made based on the hierarchical clustering of the bacterial and fungal’s 100% core microbiota (Fig. 5).

Pg. 12

Line 279 - 280: Verrucomicrobia (0.066% - 9.25%)

Please rewrite as: “Verrucomicrobia (0.66% - 9.25%)”.

Pg. 14

Line 321: Fig. 6: Soil bacterial community changed significantly due to ripening.

Suggestion: Please do insert the significant level (e.g., p-value = 0.05, or 0.01, or 0.001) and if possible, the significant value (e.g., 0.03).

Pg. 14

Line 326 - 328: Based on an investigation of the soil core, the bacterial alpha diversity was significantly lower in the preharvest samples than in the postveraison samples.

Please insert the significant level and the p-value after the significantly (eg, p < 0.05, p = 0.03). The authors should take not and insert the significant values (p = 0.03) and the significant level (p < 0.05) in their script. This will enrich the findings of this research as well as prevent readers from searching for these in the tables and figures.

Pg. 17

Line 397: Microbial biological activities such as nitrogen

Please rewrite as: “Microbiological activities such as nitrogen…………….”

Pg. 18

Line 433: …..the levels were significantly higher only in relation to the lowest layer (61-90 cm).

Please insert the significant level and/or the p-value.

Pg. 21

Line 513-514: In Mád, a relatively dry August and September is conducive to grapes achieving full biological ripeness before the onset of overripening before the berries begin to shrivel.

Please rephrase as: “In Mád the relatively dry climate in August and September is conducive for grapes achieving full biological ripeness before the onset of overripening and before the berries begin to shrivel”

Pg. 22

Line 525-526: Rhizopodaceae and Mycosphaerellaceae have been found to become more influential throughout the ripening period.

Rewrite as: “Rhizopodaceae and Mycosphaerellaceae have been found to be more influential throughout the ripening period”

Figures and Tables

Dear authors, please increase the resolution of the figures and tables. For example, Figures 2A, and 2B, Figures 7A, and 7B, Figures 9D, and 9F.

Thank you.

Johnny Kofi Awoonor

(Research Scientist: CSIR-Soil Research Institute, Kumasi. Ghana)

Reviewer #3: The problem is very nice and interesting. Soil microbes and vine production must be correlated in some points.

The several sampling point resulted a huge database with an enormous potential to give answers for questions. But the qestions should be more focused and the area of investigations should be reduced.

Current questions and aims in the manuscript:

“thoroughly characterize the soil microbiota of the historic Tokaj-Mád wine region”

It is more or less was characterized, but in this form it does not say too much. Very general. No Comparison. No conclusion.

“We aimed to retrieve the hidden microbial patterns that might be descriptive of the terroir of the Mád vineyards, including the four historic vineyards (Szent Tamás, Király, Betsek, Nyúlászó). NGS was used to describe the molecular barcodes of the historic vineyard microbiotas, which are the hallmarks of the interrelated dynamic physical and chemical properties of the vineyard soil.”

A lat of data and figures, but no answer for this question. Is it differ in any way from other soils, other vineyards soils?

“We aimed to characterize the 100% core soil microbiota and the related agrobiochemical cycles and to identify factors inducing compositional shifts. Our further aim was to investigate the multifarious relationship between the biogeochemical processes and the metagenomic functional potential of the vineyard soil by carrying out correlation analyses between the physicochemical characteristics and the functional potential of the microbiota.”

An existing significant correlation in the sampled data set not necessarily mean cause and effect relationship:

“Soil Na (both plant available and nonavailable), similar to K2O and P2O5, stimulated most of the agrobiogeochemical cycles investigated, except linoleic acid metabolism and alkaloid and anthocyanin biosynthesis.”

There is correlation. What stimulates what, it cannot be obvious according to this data matrix.

“We also wanted to investigate whether the ripening of the plant exerts any effect on the soil core bacterial and fungal microbiotas over the growing season.”

Seasonal changes can be the effect of ripening, as it can be the effect of several other things: temperature, drought, rain, weeds – where there weeds or cover plants?... It is not an established conclusion.

“Finally, we planned to investigate the ARG reservoirs of the Mád vineyards. We believe that our data will emphasize the importance of soil resistome studies and facilitate the discovery of possible ARG dissemination mechanisms.”

This is a very different question. It would worth an absolutely separated manuscript. With a more focused data analysis and discussion. In this extent it is impossible, as we did not get answers to the question.

General comments: methods, data analysis, figure legends often incomplete or not clear. The resolution of some figures are very poor, some of them unreadable.

Recommendation: this data set absolutely valuable. I am sure, a more focused question with the appropriate data analysis could results answers, but a reconsideration, a focused analysis with less data and/or factors and rewrite is necessary. In this form it is incomprehensible for the readers.

Reviewer #4: The manuscript by Remenyik and colleagues is focused on the metataxonomic analysis of several vineyards in the Tokaj region. Although of potential interest the work should be shortened. The Summary suffers from serious defects as well as the discussion in which numerous statements are made that are not supported by any reference or evidence and should be properly reviewed. Some comments to try to improve the manuscript are mentioned next.

- Major comments

Abstract should be improved. The authors begin by mentioning the fungal microbiota, with a very brief mention of the bacterial microbiota and again insist on other aspects related to the fungal microbiota. My recommendation is to give more weight in the abstract to the description of the bacterial microbiota and that there be a clearer common thread when describing the main results of the work.

Materials and Methods section. Some critical information is missed.

+ Lines 583-586. The description of the types of soils should be moved to Result section.

+ Lines 587-589. How the soils samples were collected, specially at 31-60 and 61-90 cm depth?. Authors should indicate an exact and complete methodology of soil sampling.

+ Lines 591-595. A more complete description of the methods used to analyze physico-chemical characteristics of soils should be made, including references.

+ Lines 615-617. All the primers used in the Shotgun sequencing process for both bacteria and fungi should be exactly reported.

+ How the sequencing data were processed?. Were they filtered and trimmed?. Any quality analysis was carried out?. This information is important.

Results section:

+ Lines 130 to 134: Authors indicate the number of reads identified as Bacteria, Archaea, Eukaryotes and Virus. How they identified all these taxons?. We do not have any idea about the primers used for sequencing.

+ A subsection reporting the physicochemical characteristics of each of the 10 soil plots analyzed is missed.

+ Lines 137-138. Authors mention 42 bulk soil samples, but in Materials and Methods section they indicate to have taken “Altogether 60 samples were collected from 0-30 cm, 31-60 cm, 61-90 cm depth zones. One sample consisted of 15-20 subsamples picked between vines in two rows from each zone. Sixty-two percent of the samples were topsoil samples (0-30 cm), 18% of the samples were middle soil (31-60 cm), and 20% of the samples were from the bottom (61-90 cm)”. I am totally confused. Which these 42 samples are?. Not the topsoil samples since the 62% of 60 samples are 37.2 samples (which is impossible). Please, can you clarify this point?.

+ The subsection “The soil food web of the Tokaj-Mád vineyards” leads to confusion (how is it possible that at the species level, as the authors claim, fungi do not exist in the core microbiota?; lines 248-249), does not contribute anything significant to the study and should be removed.

+ Lines 414-415. The authors state “Soil microbiota-associated antibiotic resistance genes are known environmental pollutants posing a potential threat to public health”. What evidence do the authors have to make this statement? Antibiotic resistance genes are a normal part of the genome of countless microorganisms, including the antibiotic-producing microorganisms themselves, so they cannot be considered "environmental pollutants" in any way. This phrase should be removed. Also for what is affirmed in lines 416-417 “Fungal antimicrobial compounds favor the accumulation of bacterial antibiotic-resistant factors”. Any evidence?.

Discussion section. Throughout the discussion section many claims are made that are not supported by evidence or references (i.e lines 469; lines 462-465, lines 529-530, lines 40-541and many others). A careful review of this section should be carried out in order to indicate references supporting many of the statements made.

- Minor comments

Line 48: “representing 97 of fungal operational taxonomic units”.

Line 52. “Soil core bacteria changed markedly due to ripening”: What did that change consist of?. Please, describe.

Line 139. S1 Table. There is no legend for the table that indicates which soil samples these data correspond to (I assume that they must be the 42 top soil samples but it is not clear for a general reader of the manuscript). Please, correct.

Lines 161-162. According to what it is observed in Figure 1E Streptomyces taxon is not the dominant one in Betsek sample. Please, correct.

Lines 198-200. Authors state that “In general, bacterial diversity typically exerted a positive effect, while fungal diversity showed rather a negative effect on the soil chemical parameters measured in this study”. How the authors deduce this statement?. It could be also the opposite: the physico-chemical parameters of soils exert a positive or negative effect on microbial communities. Please, change.

Lines 248-249. The statement indicating that “at the species level the 100% core microbiota did not contain fungal species” is particularly strange.

Lines 414-415. The authors state “Soil microbiota-associated antibiotic resistance genes are known environmental pollutants posing a potential threat to public health”.

Line 469. Any evidence or reference supporting this statement?.

Reviewer #5: The research article by Remenyik et al. entitled “Exploring the Interplay Between the Core Microbiota, Physicochemical Factors, Agrobiochemical Cycles and Resistome in the Soil of the Historic Tokaj Mád Wine Region characterize the soil microbiota of the historic Tokaj-Mád wine región. By NGS the authors retrieve the hidden microbial patterns that might be descriptive of the terroir of the Mád vineyards. The work reports timely and interesting data contributing to deepen current knowledge on grapevine soil microbiome. Overall, the paper was well written, and the data look quite convincing. However, some points need clarification.

The accession number(PRJNA909960) was not found in SRA.

- Section “Six clusters were identified on the basis of the physicochemical properties of the bulk soil samples”

If I understood well, you collected 42 soil samples for physicochemical analysis and 60 for sequencing. How many samples were obtained from each of the 10 sites?

About the six soils clusters (SPP), what was the criteria for defining the 6 clusters. Because looking at the tree in figure s1, more than 6 groups could be defined.

Fig. 1B. Why in the analysis were 8 of the ten vineyards? (Nyúlászó and Szilvás are missing).

L 147-148 “Principal coordinate analysis (PCoA) was performed to investigate clustering according to the soil physicochemical properties (Fig. 1C) and possible correspondence between the microbial taxonomy of the historical vineyards and the other (Fig. 1D).”

Fig. 1C, Why the in the PCoA of soil properties most of the points are in the same coordinate of the axis 1 if the soils clearly present different features. Fig. 1D, It is not clear what is the point of the figure, or what are you showing.

L 155-157 “Remarkable intra-vineyard alterations were observed in the 20 most abundant fungal genera especially in the case of Szent Tamás, Betsek and Sarkad”

Were are shown this results?

L 158-160 “…bacterial genus Bradyrhizobium which are beneficial to plant development and physiology due to the excretion of the polyhydroxylated steroid phytohormone (e.g. brassinosteroids) production and nitrogen fixation…”

Bradyrhizobium only fix nitrogen in symbiosis with leguminous plants.

L 160-162 “The root colonizing rhizobacteria Streptomyces conferring pathogen resistance dominated in Betsek.”

In Nyúlászó and Szilvás the level of Streptomyces are bigger than in Betsek (Fig.1E)

- Section Pronounced community taxonomy shifts according to soil physicochemical properties.

L 184-185 “It was shown that in terms of clades, the largest differences 185 appeared among clusters SPP2, SPP3 and SPP6.”

Where is this information?

Fig.2A, More information is needed in figure caption to interpret the tree. What are the colored circles in the cladogram?

- Section The soil food web of the Tokaj-Mád vineyards

L 234-236 “On average, Bacteria (98.61% ± 5.43%) and Archaea (1.20% ± 1.02%) dominated in the samples. Virus (0.010% ± 0.022%) were less frequent than Fungi (0.15% ± 0.21%), Algae (0.021% ± 0.08%) and Protists (0.007% ± 0.0021%).”

Protists are less abundant than virus.

Fig.4A, why the same % is represented by different size?

- Section The 100% core soil microbiota

Fig. 5

L 292-295 “The scaling is determined through referencing scientific literature. If the relative occurrence of a taxon exceeds 75%, it falls under the strong category. If it surpasses 50%, it is considered moderate. If it exceeds 25%, it is classified as weak, and if it is below 25%, it is labeled as low”.

Percentage of what, the total count of bacterial OTUs in that sample?

- Section The most remarkable taxonomic shifts were observed due to ripening of the grapevine

L 299-302 “In the case of bacteria, spectacular differences in pattern dynamics were observed in community diversity, resulting in two clusters (postveraison – cluster 1, preharvest – cluster 2) with different spatial ordinations.”

Did you did any statistical test support this conclusion?

- Section Network analyses elucidate the complex dynamics of the soil fungal interactome

Fig.7A.

L 365-367 “In all cases, the size of a node represents the degree of the OTU, reflecting the number of connections that it has to other community members in the network.

All the nodes presented the same size. What does it mean the different colors in the figure?

- Section Deciphering of the soil resistome of Tokaj-Mád vineyards

Do the authors know what type of the fertilization the different vineyards use? The animal manure could incrase the ARG.

Fig.9E.

Why the 3 showed PCoA presented the same variation in the axis 1?

- Section Materials and methods

L 638-639. “The results were presented via principal coordinates analysis (PCoA) and nonmetric multidimensional scaling (NMDS) analysis.”

The NMDS is only mentioned in this section.

L 639-640 “To depict the variance between different groups, the Adonis test together with principal coordinates analysis (PCoA) was used.”

What were the results of the test? They are not mentioned throughout the manuscript.

6. PLOS authors have the option to publish the peer review history of their article (what does this mean?). If published, this will include your full peer review and any attached files.

Reviewer #1: No

Reviewer #2: No

Reviewer #3: No

Reviewer #4: No

Reviewer #5: No

---

## [Author Response · Author response to Decision Letter 0]

6 Jan 2024

Reviewer #1:

1, 2) How the subsamples were pooled, soil or community DNA were combined? Why the percentages of top, middle and deep soil samples differ?

We have provided a clarification for the method description. Please refer to lines 657-661 for details. “Metagenomic DNA was isolated from each soil sample pool, which consisted of 15-20 sub-soil samples mixed together. These samples were collected between vines in two rows from each zone. Out of the total 60 samples, 37 (~62%) were taken from the topsoil (0-30 cm), 11 samples (~18%) from the middle soil layer (31-60 cm), and 12 samples (20%) from the bottom layer (61-90 cm). The prevalence of topsoil samples is attributed to the challenges associated with sampling at greater depths, given the presence of hard rock and a thin, fertile soil layer.”

3) What is the reason of soil sample number discrepancy mentioned in the text and S1 Fig. and Table (60 vs 42 respectively)?

Thank you for the question, we added a new sentence to clarify. See lines 666-667: “After collecting 60 soil samples, 42 were successfully evaluated through total elemental analysis.”

4) In order to correctly investigate the qualitative diversity and the community structures of the samples, rarefaction and/or rank abundance curves generation is critical. Please demonstrate/ validate that the number of OTUs are saturated in all group samples to get enough sequencing depth. 

Before investigating the sample composition, we conducted a rarefraction analysis. Three samples were dropped due to insufficient OTUs. 

5) By the way, for some reason almost all of the figures and their labels are not sharp in the pdf version of the manuscript, however downloaded tiff files were legible. 

Sorry for the poor quality. We will re-upload the figures in higher resolution.

6) Please pay more attention to the proper preparation of the figure and table legends; do not use incorrect, inconsistent or unexplained abbreviations in the figures, or such that are not common knowledge.

Thank you for the feedback. Abbreviations, such as SPP1-6 and Vy1-10, are now consistently indicated on the figures beside the captions.

7) However, it is hard to believe that “direct analysis of individual reads rather than contigs”, would provide more reliable results, especially in the case of sort reads (2 × 151 bp).

We initially ran the program in 'assembly mode,' but the assembly quality was significantly low. Subsequently, the sqm_reads program yielded much better results. Therefore, we utilized these data. 

8) As for the quantity of reads generated from the samples – 20 million reads per sample – it highly questionable that such a uniform high-quality read count could be correct. 

we added a new sentence Please refer to lines 693-696 for details “The sequencing yielded a minimum of 20 million reads per sample. To ensure the availability of 20 million reads per sample for shotgun sequencing, we achieved this by re-isolating from each sample until obtaining the required purity OD260/280=1.8-2.0 and concentration ≥ 10 ng/μL for each metagenomic isolate.”

9) Although it's a small thing, MagNA Lyser Instrument indicates oscillation frequency (speed) in revolutions per minute (rpm) not in G-force. 

Thank you for noticing, the sentence was corrected see lines 682-684 “With the use of a MagNA Lyser Instrument (Roche Applied Sciences, Germany), samples were lysed twice at 6,000 × RPM for 30 seconds each.”

10) Nevertheless, these are minor remarks, not affecting unfavorably the value of the work presented. The results and discussion section nicely follows the logic of the materials and methods. It mostly presents the findings clearly. In general, the writing style of the manuscript is proper, and the text contains only a few typographic errors and ambiguous phrasing, however personally I would like to see a much more intense and detailed result comparison with previous knowledge during the discussion.

We appreciate the observation, and in response, we have expanded the discussion, incorporating additional insights from others, compared our data with theirs, always supported by new references. All modifications have been traced in the 'marked manuscript.

Reviewer #2:

1) I suggest that the authors, if possible, should rephrase the title of the manuscript, some sentences in the manuscript and also include the significant level (e.g., 0.05, 0.01, 0.001) and the p - values (e.g., p < 0.03), if possible, into their findings where significant or not significant and/or where positive or negative corelation (r = 0.00; p -value) is reported in the manuscript.

Thank you for the suggestion. We revised the title to: “Exploring the Interplay Between the Core Microbiota, Physicochemical Factors, and Agrobiochemical Cycles in the Soil of the Historic Tokaj Mád Wine Region”. We excluded the resistome from the title and in response to the reviewer's recommendation, we omitted a substantial portion of the resistome analysis results.

Significance levels were specified as requested; please refer to lines 218, 348, 364, 382, 386, 395, 407, 411, 415, 475. 

In the chapter titled “Mutual Interactions between soil diversity and physicochemical parameters” we systematically miswrote and interchanged the p-values with the correlation coefficients. Thank you for observing this; these errors have been corrected please see lines 229-239 also 531, 534.

2) Line 83 - 84: Metabolites produced by the rhizobiome are important contributors to the quality, fragrance and distinctive personality of the products (5,6). Please replace “personality” with “quality”.

It was corrected please see line 94.

3) Line 86 - 88: The adaptation abilities of the microbiota depend on community plasticity. Network analysis-based approaches can help in better understanding the intricate nature of microbe‒microbe and microbe-community interactions. Please rephrase the above sentence as: The adaptation abilities of the microbiota depend on community plasticity. Network analysis-based approaches can help understand the intricate nature of microbe‒microbe and microbe-community interactions.

It was corrected, please see lines 96-98. “The adaptation abilities of the microbiota depend on community plasticity. Network analysis-based approaches can help understand the intricate nature of microbe-microbe and microbe-community interactions.”.

4) Line 244 - 245: At the phylum level, 90.0% of the Bacteria and 70.0% of the Fungi OTUs were present in our sample population. Please edit as: “At the phylum level, 70% and 90% of Fungi OTUs and Bacteria were present in our sample population, respectively”.

It was corrected, please see lines 274-276.

5) Line 255 - 256: Comprehensive comparisons of the soil samples of the Tokaj-Mád vineyards were made based on the hierarchical clustering of the bacterial and fungal 100% core microbiota (Fig. 5). Please rephrase the above sentence as: Comprehensive comparisons of the soil samples of the Tokaj-Mád vineyards were made based on the hierarchical clustering of the bacterial and fungal’s 100% core microbiota (Fig. 5).

It was corrected, please see lines 287-288. 

6) Line 279 - 280: Verrucomicrobia (0.066% - 9.25%). Please rewrite as: “Verrucomicrobia (0.66% - 9.25%)”. The data in the manuscript was correct.

Fig. 6: Soil bacterial community changed significantly due to ripening. We have addressed imprecise wording and adjusted the title of Figure 6 to: “Ripening had a remarkable impact on the alterations in bacterial communities” as indicated in line 358.

7 and 8) Suggestion: Please do insert the significant level (e.g., p-value = 0.05, or 0.01, or 0.001) and if possible, the significant value (e.g., 0.03). Please see lines: 327-331 and Line 346 - 348: “Based on an investigation of the soil core, the bacterial alpha diversity was significantly (p < 0.05) lower in the preharvest samples than in the postveraison samples.”

Significance levels were specified as requested; please refer to lines 218, 348, 364, 382, 386, 395, 407, 411, 415, 475. 

In the chapter titled “Mutual Interactions between soil diversity and physicochemical parameters” we systematically miswrote and interchanged the p-values with the correlation coefficients. Thank you for observing this; these errors have been corrected please see lines 229-239 also 531, 534.

9) Line 397: Microbial biological activities such as nitrogen. Please rewrite as: “Microbiological activities such as nitrogen…………….”

Thank you for the advice, it was corrected please see line: 437

10) Line 433: the levels were significantly higher only in relation to the lowest layer (61-90 cm). Please insert the significant level and/or the p-value. Thank you, we added the p-value. Please refer to line 475. 

11) Line 513-514: In Mád, a relatively dry August and September is conducive to grapes achieving full biological ripeness before the onset of overripening before the berries begin to shrivel. Please rephrase as: “In Mád the relatively dry climate in August and September is conducive for grapes achieving full biological ripeness before the onset of overripening and before the berries begin to shrivel”

Thank you for the advice, the sentence was corrected please see lines 573-577. “In Mád, the climatic conditions in August and September are relatively dry, which helps the grapes to achieve full biological ripeness before they start to overripen or shrivel. We have discovered that microbial interactions are more active during the ripening period, which leads to enhanced community modularities in postveraison and preharvest samples. A strong community safety net is represented by networks with dense connections between the nodes.”

12) Line 525-526: Rhizopodaceae and Mycosphaerellaceae have been found to become more influential throughout the ripening period. Rewrite as: “Rhizopodaceae and Mycosphaerellaceae have been found to be more influential throughout the ripening period”

Thank you, it was corrected. Please see line: 585-586.

13) Figures and Tables

Dear authors, please increase the resolution of the figures and tables. For example, Figures 2A, and 2B, Figures 7A, and 7B, Figures 9D, and 9F.

Apologies for the poor quality; the figures will be reuploaded in higher resolution.

Reviewer #3:

1) Current questions and aims in the manuscript:

“thoroughly characterize the soil microbiota of the historic Tokaj-Mád wine region”

It is more or less was characterized, but in this form it does not say too much. Very general. No Comparison. No conclusion. “We aimed to retrieve the hidden microbial patterns that might be descriptive of the terroir of the Mád vineyards, including the four historic vineyards (Szent Tamás, Király, Betsek, Nyúlászó). NGS was used to describe the molecular barcodes of the historic vineyard microbiotas, which are the hallmarks of the interrelated dynamic physical and chemical properties of the vineyard soil.” A lot of data and figures, but no answer for this question. Is it differ in any way from other soils, other vineyards soils?

We have revised the objectives in the introduction, as indicated in lines 124-136, and also compared our findings with those of others in the discussion. “Descriptive microbial studies mirror the combined effects of all physical and chemical factors enhancing our understanding of the factors influencing plant health and development. Our goal was to explore the specific microbial clades contributing to the variations among six clusters, which were differentiated based on the physicochemical characteristics of soil samples. We investigated whether soil organic and mineral content correlates more strongly with bacterial diversity than fungal diversity. We also aimed to explore the variations in the 100% bacterial and fungal microbiota of the soil core throughout the ripening stage. This process enables us to capture the cumulative effects of factors such as temperature, drought, rain, grape cultivation practices (including weeds or cover plants, etc.), and plant physiology. We focused on identifying and analyzing the taxonomic shifts within the fungal microbial interactome during the ripening of grapevines. We also aimed to investigate the relationship between bacterial and fungal phylogenetic diversity-, microbial agrobiochemical processes and soil physichochemical parameters of the vineyard soil at varying soil depths”.

Regarding the distribution of key soil bacteria, we did not observe major distortions from what is reported in the literature. However, diversities differed from values measured by others (please refer to lines 539-548 “In the case of Mádi vineyards, diversity values were found to lag behind those measured in other studies. Specifically, when compared to an Argentinian study conducted in a recently established Malbec wine region, where Shannon diversity was reported between 7-8 for bacteria and 6-8 for fungi(37). Similarly, a study investigating the soil microbiota in the Italian province of Trentino found Bacterial Shannon diversity between 6-7 and Fungal Shannon diversity between 4-6 (38) Furthermore, a German study examining the impact of soil management practices on soil microbial diversity in nine German vineyards reported Bacterial Shannon diversity ranging from 6.6-7.2 and Fungal diversity from 3.2-5(39). Moreover, in our case, fungal diversities were found to surpass the diversity values of bacteria”). We have elaborated on this in the discussion section, complementing the manuscript with three new references (37-39).

We also found the highlighted comment by both the reviewer and the corresponding section to be exaggerated. Therefore, we have removed this portion (lines: 118-123) as it did not contribute substantively to the merit of the work. In a manner previously mentioned, while the composition aligns, the diversities we measured differ from values obtained by others. 

2) “We aimed to characterize the 100% core soil microbiota and the related agrobiochemical cycles and to identify factors inducing compositional shifts. Our further aim was to investigate the multifarious relationship between the biogeochemical processes and the metagenomic functional potential of the vineyard soil by carrying out correlation analyses between the physicochemical characteristics and the functional potential of the microbiota.” An existing significant correlation in the sampled data set not necessarily mean cause and effect relationship:

As mentioned, we reformulated the objectives (see lines: 116-136), which also encompasses these sentences. We also incorporated the following sentence to the discussion section: “We explored the influence of variables including altitude, time, and sampling depth on the composition of the soil microbiomes, along with their repercussions on ecosystem services (lines: 552-554). By comparing the beta diversities of the samples collected postveraison and preharvest, we investigated the temporal dynamics of the 100% soil bacterial and fungal core microbiota of Tokaj-Mád. Please refer to lines: 552-554: “Our investigation revealed significant changes in the soil bacterial core, particularly in the two main phyla, Proteobacteria and Actinobacteria, attributed to the ripening process.”

3) “Soil Na (both plant available and nonavailable), similar to K2O and P2O5, stimulated most of the agrobiogeochemical cycles investigated, except linoleic acid metabolism and alkaloid and anthocyanin biosynthesis.” There is correlation. What stimulates what, it cannot be obvious according to this data matrix.

We corrected and replaced this sentence with this: “By investigating the correlation between the 26 biological activities and the soil physicochemical parameters, we found that soil Na (both plant available and nonavailable), similar to the K2O and P2O5 contents, stimulated most of the agrobiochemical cycles investigated, except linoleic acid metabolism and alkaloid and anthocyanin biosynthesis.” (lines: 440-444)

4) “We also wanted to investigate whether the ripening of the plant exerts any effect on the soil core bacterial and fungal microbiotas over the growing season.”

Seasonal changes can be the effect of ripening, as it can be the effect of several other things: temperature, drought, rain, weeds – where there weeds or cover plants?... It is not an established conclusion.

We corrected the sentence to this see lines: 129-132 “We also aimed to explore the variations in the bacterial and fungal microbiota of the soil core throughout the ripening stage. This process enables us to capture the cumulative effects of factors such as temperature, drought, rain, grape cultivation practices (including weeds or cover plants, etc.), and plant physiology.”

5) “Finally, we planned to investigate the ARG reservoirs of the Mád vineyards. We believe that our data will emphasize the importance of soil resistome studies and facilitate the discovery of possible ARG dissemination mechanisms.”

This is a very different question. It would worth an absolutely separated manuscript. With a more focused data analysis and discussion. In this extent it is impossible, as we did not get answers to the question.

Thank you for your observation. We also agree that our data does not support the claims made. Furthermore, we acknowledge that elaborating on these points would require a more comprehensive work. As a result, we have significantly shortened the final chapter, removing sections from the results, including a portion of Figure 9f. We retained the description of the main resistances found in the vineyards at three soil levels (including resistance classes and resistance genes corresponding to the five most common classes) both in the discussion (see erased lines: 483-489).

6) General comments: methods, data analysis, figure legends often incomplete or not clear. The resolution of some figures are very poor, some of them unreadable.

Recommendation: this data set absolutely valuable. I am sure, a more focused question with the appropriate data analysis could results answers, but a reconsideration, a focused analysis with less data and/or factors and rewrite is necessary. In this form it is incomprehensible for the readers.

We have thoroughly reviewed and clarified the methods, data analysis, and figure legends, which were often incomplete or unclear. This time, we have uploaded the figures in high resolution, ensuring readability, and we have made abbreviations easily understandable and transparent in all instances.

Reviewer #4:

The manuscript by Remenyik and colleagues is focused on the metataxonomic analysis of several vineyards in the Tokaj region. Although of potential interest the work should be shortened. The Summary suffers from serious defects as well as the discussion in which numerous statements are made that are not supported by any reference or evidence and should be properly reviewed. Throughout the discussion section many claims are made that are not supported by evidence or references (i.e lines 469; lines 462-465, lines 529-530, lines 40-541and many others). A careful review of this section should be carried out in order to indicate references supporting many of the statements made.

Thank you. Following the suggestions, we have revised the manuscript by shortening it. For enhanced clarity, we omitted the dendrogram analysis from Chapter 2, highlighting the significantly variable community components solely based on LDA scores. We significantly shortened the final chapter as well. 

However, upon careful examination of the discussion, we incorporated references as suggested to substantiate the statements. The aims of the manuscript have been rephrased; please refer to lines 116-136. Additionally, new references have been added to the discussion section; please see references 6, 31, 32, 35, 37, 38, 39, 40, 41.

Abstract should be improved. The authors begin by mentioning the fungal microbiota, with a very brief mention of the bacterial microbiota and again insist on other aspects related to the fungal microbiota. My recommendation is to give more weight in the abstract to the description of the bacterial microbiota and that there be a clearer common thread when describing the main results of the work.

We updated the abstract by removing the statement “Soil core bacteria changed markedly due to ripening” and introducing a new statement (line 60): “During ripening, the soil fungal microbiota clustered into distinct groups based on altitude, a pattern not reflected in bacteriomes”. 

Additionally, we included sentences pertaining to the description of the bacterial microbiota and its changes in relation to soil physichochemical parameters. Please refer to lines 52-62: “Linear discriminant analysis effect size was performed, revealing pronounced shifts in community taxonomy based on soil physicochemical properties. Twelve clades exhibited the most significant shifts (LDA > 4.0), including the phyla Verrucomicrobia, Bacteroidetes, Chloroflexi, and Rokubacteria, the classes Acidobacteria, Deltaproteobacteria, Gemmatimonadetes, and Betaproteobacteria, the order Sphingomonadales, Hypomicrobiales, as well as the family Sphingomonadaceae and the genus Sphingomonas. Three out of the four historic vineyards exhibited the highest occurrences of the bacterial genus Bradyrhizobium, known for its positive influence on plant development and physiology through the secretion of steroid phytohormones.Soil core bacteria changed markedly due to ripening During ripening, the soil fungal microbiota clustered into distinct groups based on altitude, a pattern not reflected in bacteriomes”.

Lines 583-586. The description of the types of soils should be moved to Result section.

Please see lines 161-163. “The main soil types of all vineyards are Luvisols (brown forest soils with clay illuviation) and Fluvisols (meadow soils with salt accumulation in deeper layers) assigned to clay loam or clay textural classes”.

Lines 587-589. How the soils samples were collected, specially at 31-60 and 61-90 cm depth?. Authors should indicate an exact and complete methodology of soil sampling.

+ Lines 591-595. A more complete description of the methods used to analyze physico-chemical characteristics of soils should be made, including references.

Thank you for the question new sentences were added. please see lines: 648-657

“Soil samples were obtained at pre-defined management plots representing typical relief, slope exposure and vineyard quality classes. Sampling has been carried out with a P4000 type truck-mounted soil sampler platform (Veris Technologies (Salina, KS, USA). Soil cores were taken using disposable PETG liners as sampler tubes; the cutting shoe was cleaned with 3% hydrogen-peroxide solution following every sampling point. Each sample, consisting of 15-20 sub-samples, was taken from two rows of each zone, between vine plants. The depth of coring varied depending on the presence of coarse textures, such as stones and rocks in the soil profile, and the depth of bedrock, resulting in different sampling depths and, therefore, different number of samples from the sampling locations.”

Lines 615-617. All the primers used in the Shotgun sequencing process for both bacteria and fungi should be exactly reported.

Please see line 689-692: The DNA was fragmented and 5' Adapter:5'-AGATCGGAAGAGCGTCGTGTAGGGAAAGAGTGTAGATCTCGGTGGTCGCCGTATCATT-3', and 3' Adapter: 5'-GATCGGAAGAGCACACGTCTGAACTCCAGTCACGGATGACTATCTCGTATGCCGTCTTCTGCTTG-3' adapters were used.

How the sequencing data were processed?. Were they filtered and trimmed?. Any quality analysis was carried out?. This information is important.

Sequencing data was not filtered. The sentence was clarified: “Shotgun sequencing data were analysed with the SqueezeMeta program, a fully automated metagenomics pipeline on default settings without trimming”. see lines: 699-700

Lines 130 to 134: Authors indicate the number of reads identified as Bacteria, Archaea, Eukaryotes and Virus. How they identified all these taxons? We do not have any idea about the primers used for sequencing.

Please see line in manuscript:

The program uses GenBank, a comprehensive database that contains publicly available nucleotide sequences of species for taxonomic profiling.

Lines 137-138. Authors mention 42 bulk soil samples, but in Materials and Methods section they indicate to have taken “Altogether 60 samples were collected from 0-30 cm, 31-60 cm, 61-90 cm depth zones. One sample consisted of 15-20 subsamples picked between vines in two rows from each zone. Sixty-two percent of the samples were topsoil samples (0-30 cm), 18% of the samples were middle soil (31-60 cm), and 20% of the samples were from the bottom (61-90 cm)”. I am totally confused. Which these 42 samples are?. Not the topsoil samples since the 62% of 60 samples are 37.2 samples (which is impossible). Please, can you clarify this point?

This section was clarified please see line: From the total of 60 samples 37 (~62%) were topsoil samples (0-30 cm), 11 samples (~18%) were middle soil (31-60 cm), and 12 samples (20%) were from the bottom (61-90 cm).

we added a new sentence to clarify. See lines 667-668: “After collecting 60 soil samples, 42 were successfully evaluated through total elemental analysis”.

How is it possible that at the species level, as the authors claim, fungi do not exist in the core microbiota?

Fungi did not have a high abundance in taxonomy. This is the explanation for that not one fungal species was presented in all of the samples. 

Lines 248-249), does not contribute anything significant to the study and should be removed.

Thank you for the advice, the sentence was removed (lines: 227-229). 

Lines 414-415. The authors state “Soil microbiota-associated antibiotic resistance genes are known environmental pollutants posing a potential threat to public health”. 

What evidence do the authors have to make this statement? Antibiotic resistance genes are a normal part of the genome of countless microorganisms, including the antibiotic-producing microorganisms themselves, so they cannot be considered "environmental pollutants" in any way. This phrase should be removed. Also for what is affirmed in lines 416-417 “Fungal antimicrobial compounds favor the accumulation of bacterial antibiotic-resistant factors”. Any evidence?. 

We appreciate the feedback; we have modified the sentence. please see lines: 110-112

Line 48: “representing 97 of fungal operational taxonomic units”. Corrected please see line: 48

Line 52. “Soil core bacteria changed markedly due to ripening”: What did that change consist of?. Please, describe.

We omitted the indicated sentence and replaced it with a new one: 'During ripening, the taxonomical composition of the soil fungal microbiota clustered into distinct groups depending on altitude, differences that were not reflected in bacteriomes.' See lines: 60-62.

Line 139. S1 Table. There is no legend for the table that indicates which soil samples these data correspond to (I assume that they must be the 42 top soil samples but it is not clear for a general reader of the manuscript). Please, correct.

Both the table and the legend have been supplemented with the missing data: “Organic and mineral content analysis was conducted on 42 soil samples collected from vineyards in Mád. The origin of each sample was specified, indicating the vineyard plot and soil depth”. Please see line: 927-929 

Lines 161-162. According to what it is observed in Figure 1E Streptomyces taxon is not the dominant one in Betsek sample. Please, correct.

Thank you for noticing, the sentence was corrected please see lines: 186-187

‘The root colonizing rhizobacteria Streptomyces conferring pathogen resistance dominated in Nyúlászó and Szilvás’

Lines 198-200. Authors state that “In general, bacterial diversity typically exerted a positive effect, while fungal diversity showed rather a negative effect on the soil chemical parameters measured in this study”. How the authors deduce this statement?. It could be also the opposite: the physico-chemical parameters of soils exert a positive or negative effect on microbial communities. Please, change.

We appreciate the correction; it has been changed “In general, the physico-chemical parameters of soils can either positively or negatively affect microbial communities”, pls. see lines: 226-229.

Lines 248-249. The statement indicating that “at the species level the 100% core microbiota did not contain fungal species” is particularly strange.

Fungi did not have a high abundance in taxonomy. This is the explanation for that not one fungal species was presented in all of the samples.

Lines 414-415. The authors state “Soil microbiota-associated antibiotic resistance genes are known environmental pollutants posing a potential threat to public health”. Line 469. Any evidence or reference supporting this statement?

We appreciate the observation; we have deleted the statement (455-456).

Reviewer #5:

The accession number (PRJNA909960) was not found in SRA.

The sequencing data was uploaded but it was not released. Please check this reviewer link. The data will be released when the article is accepted.

https://dataview.ncbi.nlm.nih.gov/object/PRJNA909960?reviewer=de9gvqaubov0ud38dbbtb284al

If I understood well, you collected 42 soil samples for physicochemical analysis and 60 for sequencing. How many samples were obtained from each of the 10 sites?

Yes, and we have enhanced Table S1 by specifying the origin of each sample, indicating the vineyard plot and soil depth. This allows for the identification of the number of samples collected from each specific plot. Lines: 927-929 “Organic and mineral content analysis was conducted on 42 soil samples collected from vineyards in Mád. The origin of each sample was specified, indicating the vineyard plot and soil depth”

About the six soils clusters (SPP), what was the criteria for defining the 6 clusters. Because looking at the tree in figure s1, more than 6 groups could be defined.

Thank you, indeed, it could have been possible to form more clusters; however, during the clustering process, we aimed to rationalize the number based on the size of the sampled population. The goal was to have approximately an equal number of samples in each cluster. 

Fig. 1B. Why in the analysis were 8 of the ten vineyards? (Nyúlászó and Szilvás are missing).

 Through physicochemical analysis we did not managed to analyse samples from Nyúlászó and Szilvás. The labels are removed from the figure.

L 147-148 “Principal coordinate analysis (PCoA) was performed to investigate clustering according to the soil physicochemical properties (Fig. 1C) and possible correspondence between the microbial taxonomy of the historical vineyards and the other (Fig. 1D).” Fig. 1C, Why the in the PCoA of soil properties most of the points are in the same coordinate of the axis 1 if the soils clearly present different features. Fig. 1D, It is not clear what is the point of the figure, or what are you showing.

In the case of Figure D, we would like to emphasize that, concerning the taxonomic composition of soils, the points representing the highlighted vineyards are closely situated, indicating a similarity based on taxonomy. However, compared to other vineyards, there are significant differences. With this, our goal is to demonstrate that the microbial fingerprints of the highlighted vineyards could be distinguished from those of the other areas. We also added a sentence to the discussion please see lines: 522-524. “Concerning the taxonomic composition of vineyard soils, it can be estimated that microbial fingerprints of the historic vineyards are distinguishable from those of other areas in the Tokaj-Mád wine region, a distinction that arises from the exhibited taxonomic similarities.”

L 155-157 “Remarkable intra-vineyard alterations were observed in the 20 most abundant fungal genera especially in the case of Szent Tamás, Betsek and Sarkad” Were are shown this results?

Yes, please see Fig1/F. Szent Tamás, Betsek and Sarkad Fungal compositions are remarkably different from other vineyards.

L 158-160 “…bacterial genus Bradyrhizobium which are beneficial to plant development and physiology due to the excretion of the polyhydroxylated steroid phytohormone (e.g. brassinosteroids) production and nitrogen fixation…”

Bradyrhizobium only fix nitrogen in symbiosis with leguminous plants.

Thank you, the nitrogen fixation statement was removed. Please see line: 185

L 160-162 “The root colonizing rhizobacteria Streptomyces conferring pathogen resistance dominated in Betsek.” In Nyúlászó and Szilvás the level of Streptomyces are bigger than in Betsek (Fig.1E)

Thank you for noticing, the stement was corrected, please see lines: 185-187. “The root colonizing rhizobacteria Streptomyces conferring pathogen resistance dominated in Nyúlászó and Szilvás”.

L 184-185 “It was shown that in terms of clades, the largest differences 185 appeared among clusters SPP2, SPP3 and SPP6.”

Where is this information? 

Please see Figure2A, the highest number of taxa involved in significant taxonomic shifts are in clusters SPP2, SPP3 and SPP6.

Fig.2A, More information is needed in figure caption to interpret the tree. What are the colored circles in the cladogram?

Another reviewer was also troubled by the cladogram, so we removed it from the manuscript and from the Figure 2A, aiming to facilitate better understanding.

L 234-236 “On average, Bacteria (98.61% ± 5.43%) and Archaea (1.20% ± 1.02%) dominated in the samples. Virus (0.010% ± 0.022%) were less frequent than Fungi (0.15% ± 0.21%), Algae (0.021% ± 0.08%) and Protists (0.007% ± 0.0021%).”

Protists are less abundant than virus.

Thank you for noticing, it was corrected please see line: 265-266

On average, Bacteria (98.61% ± 5.43%) and Archaea (1.20% ± 1.02%) dominated in the samples. Virus (0.010% ± 0.022%) were less frequent than Fungi (0.15% ± 0.21%), Algae (0.021% ± 0.08%). Protists (0.007% ± 0.0021%) were the less frequent component.

Fig.4A, why the same % is represented by different size?

In the case of this figure, we partitioned the bar percentages (0-0.06, than 0.07-5, than 5-100) because the small numbers would not have been otherwise discernible. 

L 292-295 “The scaling is determined through referencing scientific literature. If the relative occurrence of a taxon exceeds 75%, it falls under the strong category. If it surpasses 50%, it is considered moderate. If it exceeds 25%, it is classified as weak, and if it is below 25%, it is labeled as low”. Percentage of what, the total count of bacterial OTUs in that sample?

Thank you for the question, the statement was clarified, please see line: 327-330.

“The taxons of ecosystem services are a total 100%. If the relative occurrence of an ecosystem service exceeds 75%, it falls under the strong category. If it surpasses 50%, it is considered moderate. If it exceeds 25%, it is classified as weak, and if it is below 25%, it is labelled as low.”

L 299-302 “In the case of bacteria, spectacular differences in pattern dynamics were observed in community diversity, resulting in two clusters (postveraison – cluster 1, preharvest – cluster 2) with different spatial ordinations.”

Did you did any statistical test support this conclusion?

No, this is based on the clusters in PCoA plot.

L 365-367 “In all cases, the size of a node represents the degree of the OTU, reflecting the number of connections that it has to other community members in the network.

All the nodes presented the same size. What does it mean the different colors in the figure?

Thank you for the question. The figure description was clarified, please see lines: 403-405

“Data are normalized, green edges correspond to positive correlations and red edges to negative ones. Node colours represent clusters, which are determined using hierarchical clustering.”

Do the authors know what type of the fertilization the different vineyards use? The animal manure could incrase the ARG.

This aspect was not investigated, thank you for the advice.

Fig.9E. Why the 3 showed PCoA presented the same variation in the axis 1?

We have condensed this chapter, removing Figure 9F and parts of Figure 9E. We retained only the results for the 0-30 cm depth in the case of the PCoA.

- Section Materials and methods

L 638-639. “The results were presented via principal coordinates analysis (PCoA) and nonmetric multidimensional scaling (NMDS) analysis.”

The NMDS is only mentioned in this section.

Thank you for the suggestion, the NMDS statement was removed. Lines: 717-719

L 639-640 “To depict the variance between different groups, the Adonis test together with principal coordinates analysis (PCoA) was used.”

What were the results of the test? They are not mentioned throughout the manuscript.

Thank you for noticing, the sentence was removed. Lines: 717-719

---

## [Decision Letter · Decision Letter 1]

16 Feb 2024

PONE-D-23-21640R1Exploring the Interplay Between the Core Microbiota, Physicochemical Factors, Agrobiochemical Cycles in the Soil of the Historic Tokaj Mád Wine RegionPLOS ONE

Dear Dr. Paholcsek,

Thank you for submitting your manuscript to PLOS ONE. After careful consideration, we feel that it has merit but does not fully meet PLOS ONE’s publication criteria as it currently stands. Therefore, we invite you to submit a revised version of the manuscript that addresses the points raised during the review process.

**The changes made in addressing the reviewers comments are appreciated and have resulted in a manuscript that is close to meeting the threshold for acceptance. Please closely consider the comments from reviewer 5 and address them in a revised manuscript.**

We look forward to receiving your revised manuscript.

Kind regards,

Theodore Raymond Muth

Academic Editor

PLOS ONE

Journal Requirements:

Reviewers' comments:

Reviewer's Responses to Questions

**Comments to the Author**

1. If the authors have adequately addressed your comments raised in a previous round of review and you feel that this manuscript is now acceptable for publication, you may indicate that here to bypass the “Comments to the Author” section, enter your conflict of interest statement in the “Confidential to Editor” section, and submit your "Accept" recommendation.

Reviewer #1: All comments have been addressed

Reviewer #2: All comments have been addressed

Reviewer #5: (No Response)

2. Is the manuscript technically sound, and do the data support the conclusions?

Reviewer #1: Yes

Reviewer #2: Yes

Reviewer #5: Yes

3. Has the statistical analysis been performed appropriately and rigorously? 

Reviewer #1: Yes

Reviewer #2: Yes

Reviewer #5: No

4. Have the authors made all data underlying the findings in their manuscript fully available?

Reviewer #1: Yes

Reviewer #2: Yes

Reviewer #5: Yes

5. Is the manuscript presented in an intelligible fashion and written in standard English?

Reviewer #1: Yes

Reviewer #2: Yes

Reviewer #5: Yes

6. Review Comments to the Author

Reviewer #1: The authors satisfactorily answered the questions and complied with the requested changes, so I consider the revised version of the manuscript suitable for publication.

Best regards

Reviewer #2: The authors have done a great job on the subject matter and I wish them all the best in this field of study.

Reviewer #5: On reading through the "new version” of the manuscript, I was made aware of the many changes that have been made explain the observed points. But not all concerns reported in my previous evaluation report have been addressed carefully.

L 147-148 “Principal coordinate analysis (PCoA) was performed to investigate clustering according to the soil physicochemical properties (Fig. 1C) and possible correspondence between the microbial taxonomy of the historical vineyards and the other (Fig. 1D).” Fig. 1C, Why the in the PCoA of soil properties most of the points are in the same coordinate of the axis 1 if the soils clearly present different features. Fig. 1D, It is not clear what is the point of the figure, or what are you showing.

In the case of Figure D, we would like to emphasize that, concerning the taxonomic composition of soils, the points representing the highlighted vineyards are closely situated, indicating a similarity based on taxonomy. However, compared to other vineyards, there are significant differences. With this, our goal is to demonstrate that the microbial fingerprints of the highlighted vineyards could be distinguished from those of the other areas. We also added a sentence to the discussion please see lines: 522-524. “Concerning the taxonomic composition of vineyard soils, it can be estimated that microbial fingerprints of the historic vineyards are distinguishable from those of other areas in the Tokaj-Mád wine region, a distinction that arises from the exhibited taxonomic similarities.”

The authors didn’t answer my question about the Figure 1C.

L 155-157 “Remarkable intra-vineyard alterations were observed in the 20 most abundant fungal genera especially in the case of Szent Tamás, Betsek and Sarkad” Were are shown this results?

Yes, please see Fig1/F. Szent Tamás, Betsek and Sarkad Fungal compositions are remarkably different from other vineyards.

What you can see in figure 1f is the inter-vineyards variation and not intra-vineyards

L 299-302 “In the case of bacteria, spectacular differences in pattern dynamics were observed in community diversity, resulting in two clusters (postveraison – cluster 1, preharvest – cluster 2) with different spatial ordinations.” Did you did any statistical test support this conclusion?

No, this is based on the clusters in PCoA plot.

In PCoA plot to define valid cluster a statistical test should be done

7. PLOS authors have the option to publish the peer review history of their article (what does this mean?). If published, this will include your full peer review and any attached files.

Reviewer #1: No

Reviewer #2: No

Reviewer #5: No

---

## [Author Response · Author response to Decision Letter 1]

22 Feb 2024

L 147-148 “Principal coordinate analysis (PCoA) was performed to investigate clustering according to the soil physicochemical properties (Fig. 1C) and possible correspondence between the microbial taxonomy of the historical vineyards and the other (Fig. 1D).” Fig. 1C, Why the in the PCoA of soil properties most of the points are in the same coordinate of the axis 1 if the soils clearly present different features. Fig. 1D, It is not clear what is the point of the figure, or what are you showing.

In the case of Figure D, we would like to emphasize that, concerning the taxonomic composition of soils, the points representing the highlighted vineyards are closely situated, indicating a similarity based on taxonomy. However, compared to other vineyards, there are significant differences. With this, our goal is to demonstrate that the microbial fingerprints of the highlighted vineyards could be distinguished from those of the other areas. We also added a sentence to the discussion please see lines: 522-524. “Concerning the taxonomic composition of vineyard soils, it can be estimated that microbial fingerprints of the historic vineyards are distinguishable from those of other areas in the Tokaj-Mád wine region, a distinction that arises from the exhibited taxonomic similarities.”

The authors didn’t answer my question about the Figure 1C.

Thank you for the question. In the case of Fig1C, the Principal Coordinate Analysis (PCoA) was performed, clustering according to the soil physicochemical properties. Here, a different dataset was used compared to Fig 1D, which is based on taxonomic data. The complexity of metagenomic datasets is typically much higher than that of data related to soil physicochemical parameters. This difference in complexity might also contribute to the variations observed between the two figures, as metagenomic data encompass a broader spectrum of biological information that can influence the clustering patterns observed in the PCoA.

L 155-157 “Remarkable intra-vineyard alterations were observed in the 20 most abundant fungal genera especially in the case of Szent Tamás, Betsek and Sarkad” Were are shown this results?

Yes, please see Fig1/F. Szent Tamás, Betsek and Sarkad Fungal compositions are remarkably different from other vineyards.

What you can see in figure 1f is the inter-vineyards variation and not intra-vineyards

Thank you for noticing. The statement was changed to: Remarkable inter-vineyard alterations were observed in the 20 most abundant fungal genera especially in the case of Szent Tamás, Betsek and Sarkad. Please see line 168- 169

L 299-302 “In the case of bacteria, spectacular differences in pattern dynamics were observed in community diversity, resulting in two clusters (postveraison – cluster 1, preharvest – cluster 2) with different spatial ordinations.” Did you did any statistical test support this conclusion?

No, this is based on the clusters in PCoA plot.

In PCoA plot to define valid cluster a statistical test should be done

Thank you for the question. The ANOSIM statistical test was employed to calculate the statistics for the Principal Coordinate Analysis (PCoA). As indicated on lines 340-341, for bacteria, distances revealed two clusters with significantly different taxonomic compositions (p = 0.01), whereas fungi displayed overlapping clusters, suggesting more similar taxonomic compositions (p = 0.165).

Furthermore, we have included a statement in the materials and methods section under Statistical Analysis, with a reference, as seen on lines 686-688: "For the statistical analysis of Principal Coordinate Analysis (PCoA), the ANOSIM permutation test was utilized to evaluate the statistical significance of the separation among groups, using the R 'vegan' package (57)."

For reference, please see lines 893-895: Cho EJ, Leem S, Kim SA, Yang J, Lee YB, Kim SS, et al. "Circulating Microbiota-Based Metagenomic Signature for Detection of Hepatocellular Carcinoma." Sci Rep. 2019 May 17;9(1):7536.

---

## [Editor Report · Decision Letter 2]

1 Mar 2024

Exploring the Interplay Between the Core Microbiota, Physicochemical Factors, Agrobiochemical Cycles in the Soil of the Historic Tokaj Mád Wine Region

PONE-D-23-21640R2

Dear Dr. Paholcsek,

We’re pleased to inform you that your manuscript has been judged scientifically suitable for publication and will be formally accepted for publication once it meets all outstanding technical requirements.

Kind regards,

Theodore Raymond Muth

Academic Editor

PLOS ONE
---

## [Editor Report · Acceptance letter]

2 Apr 2024

PONE-D-23-21640R2 

PLOS ONE

Dear Dr. Paholcsek, 

I'm pleased to inform you that your manuscript has been deemed suitable for publication in PLOS ONE. Congratulations! Your manuscript is now being handed over to our production team.

Kind regards, 

on behalf of

Dr. Theodore Raymond Muth 

Academic Editor

PLOS ONE